# Development of quasi-solid-state anode-free high-energy lithium sulfide-based batteries

Yuzhao Liu[1], Xiangyu Meng[1], Zhiyu Wang ⬡[1,2,3] ✉ & Jieshan Qiu ⬡[1,4]

Anode-free lithium batteries without lithium metal excess are a practical option to maximize the energy content beyond the conventional design of Li-ion and Li metal batteries. However, their performance and reliability are still limited by using low-capacity oxygen-releasing intercalation cathodes and flammable liquid electrolytes. Herein, we propose quasi-solid-state anode-free batteries containing lithium sulfide-based cathodes and non-flammable polymeric gel electrolytes. Such batteries exhibit an energy density of 1323 Wh L$^{-1}$ at the pouch cell level. Moreover, the lithium sulfide-based anode-free cell chemistry endows intrinsic safety thanks to a lack of uncontrolled exothermic reactions of reactive oxygen and excess Li inventory. Furthermore, the non-flammable gel electrolyte, developed from MXene-doped fluorinated polymer, inhibits polysulfide shuttling, hinders Li dendrite formation and further secures cell safety. Finally, we demonstrate the improved cell safety against mechanical, electrical and thermal abuses.

Electrification of the transportation sector is vital to realizing carbon-neutral economics. With limited energy density, however, the state-of-the-art Li-ion batteries (LIBs) are difficult to afford continuously high electricity consumption of long-range transportation. Employing Li metal anode with high theoretical capacity can potentially multiply several folds the energy of rechargeable batteries. But the use of thick Li metal anode (>500 μm in thickness) induces far excess Li, which largely offsets the actual specific energy of Li metal batteries (LMBs) and raises the cell cost[1]. For example, a 200% overuse of Li metal anode can reduce the energy density of LMBs (687 mAh L$^{-1}$) to even lower than that of LIBs with fully lithiated graphite anode (719 mAh L$^{-1}$)[2]. The thick Li metal anode also depletes the lean electrolyte or works with excess electrolyte to create a misguided impression of cell performance. Decreasing the amount of Li metal anode by reducing its thickness (<50 μm) is desired for full exploitation of high energy of LMBs. However, the fabrication of thin Li metal with high reactivity and strong adhesion property significantly raises the processing complexity and cost (> US$1000 kg$^{-1}$)[3]. Moreover, the practical potential of LMBs is also restricted by not only rapid cell failure via Li dendrite growth but also safety hazards from uncontrolled exothermic

reactions between reactive Li metal and flammable organic liquid electrolytes[4-6]. These difficulties significantly reduce the cost-effectiveness, reliability and sustainability of LMBs in the life cycle of manufacturing, storage, daily use and recycling.

Recently, anode-free batteries have emerged as a cost-effective power source[7]. Such cell design includes only Li-rich cathodes against the bare metal current collector without active anode initially. Upon charge, the active Li is released from Li-rich cathodes and deposited on the countered metal current collector as the temporal anode, allowing the subsequent discharge by similar chemistry of LMBs. This feature is advantageous in maximizing the specific energy and energy density of the batteries due to zero excess of Li metal anode. The absence of anode also enables cell production at lower processing complexity and cost. In anode-free cells, there are no extra Li sources to replenish the irreversible Li loss during cycling[8]. Therefore, the cathode with as high as Li content is vital to maintain the cell reversibility for realizing high energy. Common intercalation-type cathodes are difficult to meet this desire owing to low Li content (e.g., 14.3 at.% for LiFePO$_4$, 25 at.% for LiCoO$_2$ and LiNi$_x$Co$_y$Mn$_{1-x-y}$O$_2$) and lithium storage capacities despite the good compatibility with the existing production infrastructure of

[1]State Key Laboratory of Fine Chemicals, Liaoning Key Laboratory for Energy Materials and Chemical Engineering, Dalian University of Technology, Dalian 116024, PR China. [2]Branch of New Material Development, Valiant Co. Ltd, Yantai 265503, PR China. [3]State Key Laboratory of Organic-Inorganic Composites, Beijing University of Chemical Technology, Beijing 100029, PR China. [4]College of Chemical Engineering, Beijing University of Chemical Technology, Beijing 100029, PR China. ✉e-mail: zywang@dlut.edu.cn

LIBs. The release of reactive oxygen radicals from such oxide cathodes may also damage cell safety by triggering hazardous side reactions with flammable organic electrolytes under abuse conditions[9]. So far, the development of high-energy and safe anode-free cells remains a significant challenge due to a lack of Li-rich cathode with high capacities and satisfying reliability.

Lithium sulfide ($Li_2S$) with as high as 66.7 at% Li represents a good candidate of Li-rich cathodes for developing anode-free cells with high energy. With a fully lithiated structure, the $Li_2S$ experiences negligible volume expansion upon cycling, particularly suitable for manufacturing high-loading cathode and solid-state cells[10]. They can deliver a specific capacity of 1166 mAh g$^{-1}$ via multi-electron-involved redox chemistry beyond the intercalation cathodes[11]. This merit would be fully exerted in an anode-free cell design to acquire remarkable specific energy of 2451 Wh kg$^{-1}$ and energy density up to 4068 Wh L$^{-1}$. The oxygen-free composition of $Li_2S$ further reduces the side reactions with organic electrolytes to enhance cell reliability[12]. Nevertheless, the $Li_2S$ cathodes based on bulk $Li_2S$ generally suffer rather high initial activation overpotential (~1.0 V) and sluggish charge kinetics caused by its insulating ionic lattice and poor solubility in organic electrolyte[13]. The nanosize effect, electrocatalysts or redox mediators (e.g., $Na_2S$, quinone) have been demonstrated to be effective in reducing the redox difficulties in running $Li_2S$ cathode with low mass loading[14,15]. In principle, the anode-free cells using $Li_2S$ cathode may follow a similar pathway with Li-S batteries involving soluble lithium polysulfides (LiPS) as the redox intermediates after initial activation that stoichiometrically convert the $Li_2S$ to sulfur and Li metal on the cathode and anode side, respectively. The irreversible LiPS leakage to the electrolyte and their shuttling to contaminate the anode would damage the cell reversibility and lifetime[16]. Conventional strategies such as chemical adsorption, physical trapping and repulsive interlayer can suppress LiPS loss but is hard to eliminate it due to high LiPS solubility in organic electrolyte[17-23]. Replacing the liquid electrolyte with solid-state electrolytes provides the ultimate solution to this problem at a cost of the huge loss of interfacial compatibility and ionic transport kinetics[24-26]. The quasi-solid-state electrolytes consisting of liquid electrolyte in solid matrix offers a good compromise in ionic conductivity and interfacial properties but much better stability and LiPS blocking ability than liquid ones[27]. Applying them to strengthen Li-rich $Li_2S$ cathode is anticipated to satisfy both high cell energy and safety of anode-free cells, which, however, has been rarely explored.

In this work, we report a quasi-solid-state anode-free cell with high energy and reliability enabled by applying Li-rich, oxygen-free $Li_2S$ cathode in a robust composite gel polymer electrolyte (CGPE) with fast ion transport, good thermal stability and fire retardance (Fig. 1). The redox activity of bulk $Li_2S$ is boosted to minimize the activation and charge difficulties by cold pressing into the conductive matrix of MXene full of electron-withdrawing sites. This method can yield dense

$Li_2S$ cathodes (denoted as $Li_2S@MX$) with high $Li_2S$ loading up to 14 mg cm$^{-2}$ and high areal capacity loading. It is hardly achieved by the common slurry-casting method due to the huge resistance of insulating $Li_2S$ in thick electrodes. Such capacity loading level is comparable to present LIBs with areal mass loading of 20–30 mg cm$^{-2}$. On this basis, a robust and fireproof CGPE with poor LiPS solubility is designed to hinder not only Li dendrite growth but also LiPS shuttling while enhancing cell safety in terms of a composite of fluorinated polymer and MXene. Assembling $Li_2S@MX$ cathode in CGPE enables the quasi-solid-state anode-free cells with good reversibility and high energy density. Such cells also exhibit low self-discharge and high reliability under extreme abuse conditions thank to a synergy of stable redox chemistry of $Li_2S$ in robust and fireproof CGPE.

## Results
### Fabrication and characterization of $Li_2S@MX$ cathode
Free-standing $Li_2S@MX$ cathodes with tunable $Li_2S$ loading of 5.0–14.6 mg cm$^{-2}$ are prepared by cold pressing a mixture of bulk $Li_2S$ particles and $Ti_3C_2T_x$ MXene nanosheets at a constant pressure of 300 MPa at $25 \pm 1\,°C$. This method avoids the use of inactive auxiliary additives and heavy metal current collectors to maximize the capacity loading and allows easier electrode manufacturing. The $Li_2S$ particles with micrometer size are uniformly decorated within a conductive matrix of MXene nanosheets, yielding a dense plate with tens of micrometer thickness (Fig. 2A). X-ray powder diffraction (XRD) analysis reveals the co-existence of $Li_2S$ (JCPDS No. 23-0369) and $Ti_3C_2T_x$ MXene in this cathode (Supplementary Fig. 1). X-ray photoelectron spectroscopy (XPS) survey scan confirms the presence of Li, S and Ti elements from $Li_2S$ and $Ti_3C_2T_x$ MXene, respectively (Supplementary Fig. 2). Compared to pure $Li_2S$, the Li 1 s peak of $Li_2S@MX$ is sifted by 0.11 eV towards higher binding energy, implying a chemical interaction between $Li_2S$ and MXene (Fig. 2B). The $^7Li$ spectrum of solid-state magic-angle-spinning nuclear magnetic resonance (MAS-NMR) for $Li_2S@MX$ also shows the drift of chemical shift by ca. 0.4 ppm toward the lower field with respect to pure $Li_2S$ (Fig. 2C). This phenomenon suggests the electronic deshielding of Li atoms in $Li_2S$ by coordinating with electron-withdrawing groups on MXene surface. Taking $Li_2S$ and oxygen-functionalized $Ti_3C_2$ as the model system, the interaction between $Li_2S$ and MXene is investigated by first-principle calculation (Fig. 2D). It reveals the intimate binding of $Li_2S$ on the (001) facet of $Ti_3C_2O_2$ via Li-O bonding with high binding energy ($E_b$) of −3.93 eV. Such a strong chemical interaction expends the Li-S bond length by 14.3 % relative to the free $Li_2S$ molecule. It leads to a red-shift of $T_{2g}$ band associated with Li-S bond vibration by 5 cm$^{-1}$ in Raman spectra (Fig. 2E). Such an activation effect is beneficial to triggering the $Li_2S$ activation by weakening the Li-S bonds on $Li_2S$/MXene interface upon electrochemical oxidation, which allows the Li$^+$ extraction from $Li_2S$ with a lower energy barrier relative to nonpolar carbon (Fig. 2F). Moreover, the presence of oxygen-containing groups on MXene surface also enables strong adsorption of LiPS (e.g., $Li_2S_4$, $Li_2S_6$) with high $E_b$ of −1.89 to −2.62 eV, suggesting a good capability of suppressing LiPS leakage. This benefit may enable not only better electrode reversibility but also a high LiPS accumulation on cathode interface for propelling their redox conversion forward with ease (Fig. 2D)[28].

### Electrochemical characterizations of Li||$Li_2S@MX$ cells with non-aqueous liquid electrolyte
A $Li_2S@MX$ cathode with $Li_2S$ loading of ca. 5 mg cm$^{-2}$ is used to evaluate the intrinsic performance in ether-based liquid electrolyte against Li metal anode. All the voltages refer to Li/Li$^+$ in half cells. The specific capacities are calculated by the mass of $Li_2S$. For comparison, a control cathode with similar $Li_2S$ loading is made from a composite of bulk $Li_2S$ and carbon black ($Li_2S@C$). The bulk $Li_2S$ in $Li_2S@MX$ can be activated at a potential (2.38 V) rather close to its thermodynamic oxidation potential (2.3 V) (Fig. 3A). Compared to $Li_2S@C$ cathode, the

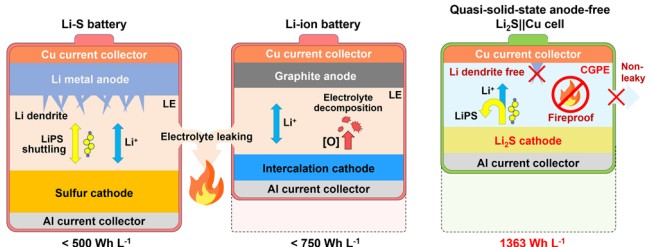

**Fig. 1 | Schematic illustration of the quasi-solid-state Li2S||Cu cells.** Schematic configurations and advantages of quasi-solid-state Li2S||Cu cells in satisfying both high energy and reliability by stable redox chemistry in robust and fire-retardant gel polymer electrolyte. (the LE, CGEP, LiPS and [O] refer to the liquid electrolyte, composite gel polymer electrolyte, lithium polysulfides and radical oxygen, respectively.).

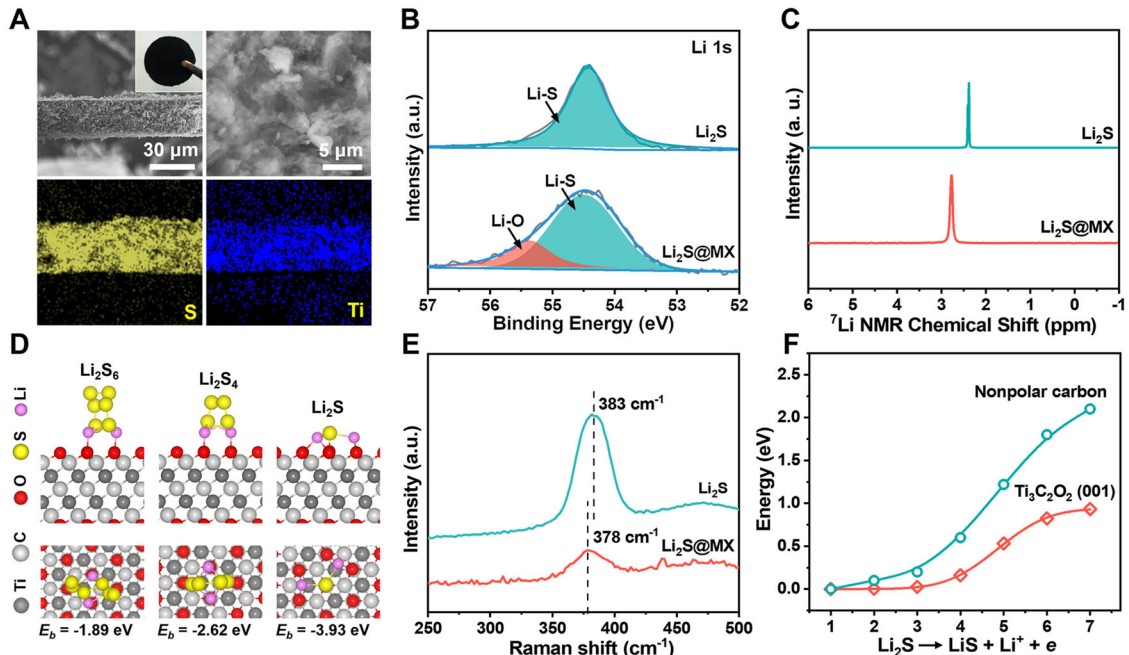

**Fig. 2 | Physicochemical characterizations of Li2S@MX cathode. A** Cross-section SEM image and elemental mapping of 2D-compacted Li2S@MX electrode, the inset of the upper left figure is an optical image of this electrode. **B** The Li 1 s XPS spectrum of pristine Li2S and Li2S@MX. **C** The solid-state $^7$Li MAS-NMR spectrum of pristine Li2S and Li2S@MX. **D** Atomic structures of Li2S, Li2S4 and Li2S6 cluster adsorbed on (001) facet of Ti3C2O2. **E** The Raman spectra of pristine Li2S and Li2S@MX. **F** Energy profiles for Li2S dissociation (Li2S → LiS + Li$^+$ + $e$) on (001) facet of Ti3C2O2 or nonpolar carbon.

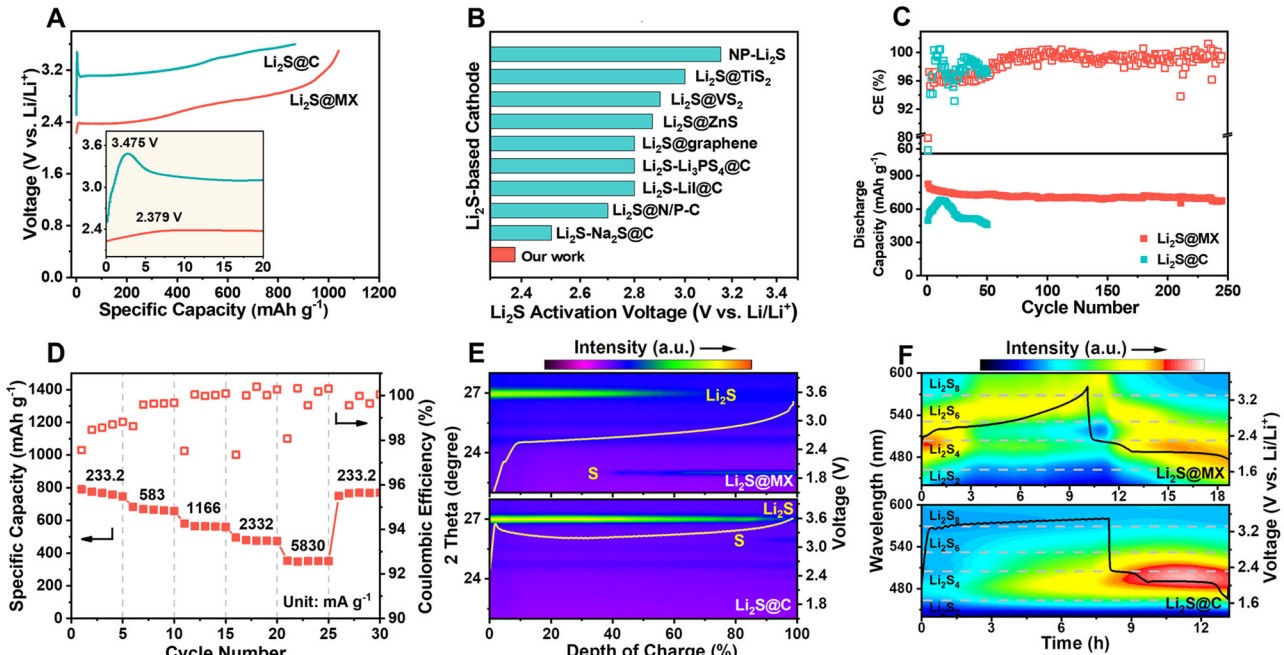

**Fig. 3 | Electrochemical and physicochemical characterizations of Li‖Li2S@MX cells with non-aqueous liquid electrolyte. A** Charge voltage curves of Li2S@MX and Li2S@C cathodes for initial activation at a specific current of 116.6 mA g$^{-1}$, the inset is the magnified view showing activation barrier of Li2S. **B** A comparison of Li2S@MX cathode and reported Li2S-based cathodes in the activation potential barrier[14,29–36]. **C** Cycling stability of Li2S@MX and Li2S@C cathodes at a specific current of 233.2 mA g$^{-1}$. **D** Rate performance of the Li2S@MX cathode at various specific currents ranging from 233.2 to 5830 mA g$^{-1}$. **E** Operando XRD contour patterns of Li2S@MX and Li2S@C cathodes upon initial charge. **F** Operando UV-vis contour patterns of Li2S@MX and Li2S@C cathodes upon cycling.

activation potential barrier of Li2S@MX is dramatically reduced by as much as 1.095 V, which is lower than most Li2S cathodes made of nanosized Li2S (Fig. 3B)[14,29–36]. Potentiostatic charge test further validates better redox activity of Li2S@MX relative to Li2S@C by much higher redox current and dissociation capacity of initial activation at 2.4 V (Supplementary Fig. 3). After initial activation, the cyclic voltammetries (CVs) of Li2S@MX cathode show stronger redox peaks with narrower voltage gaps relative to Li2S@C (Supplementary Fig. 4). Specifically, the anodic peak associated with Li2S and LiPS oxidation to sulfur is shifted to lower voltage by 300 mV while the reverse process

moves the cathodic peaks oppositely by 96–190 mV. These results suggest improved redox activity of bulk $Li_2S$ throughout the charge-discharge process in the presence of MXene.

High redox activity of Li₂S@MX cathode enables nearly full utilization of $Li_2S$ to deliver a high capacity of 1040 mAh g⁻¹ upon charge to 3.5 V for initial activation at a specific current of 116.6 mA g⁻¹ (Fig. 3C). Afterwards, a high capacity of 824 mAh g⁻¹ is achieved between 1.7–2.8 V at 233.2 mA g⁻¹. The discharge curves of this cathode show two voltage plateaus at ca. 2.3 and 2.1 V, corresponding to the generation of long-chain LiPS ($Li_2S_x$, $6 \le x \le 8$) and their conversion to short-chain ones ($Li_2S_x$, $2 < x \le 4$) and insoluble $Li_2S$, respectively (Supplementary Fig. 5)[12]. The Li₂S@MX cathode with various $Li_2S$ loading of ca. 1.0–5.0 mg cm⁻² can retain over 81–90% of initial capacity with nearly 100 % Coulombic efficiency (CE) after 250 cycles at a specific current of 233.2 mA g⁻¹ (Fig. 3C and Supplementary Fig. 6). During cycling, the discharge-charge gap ($\Delta E = 170$ mV) keeps narrow constantly, suggesting the long-term effectiveness of MXene in reducing the redox difficulty of $Li_2S$ conversion and electrode polarization (Supplementary Fig. 5). At higher specific current of 583–5830 mA g⁻¹, the Li₂S@MX cathode still manifests fast redox kinetics, delivering high capacities of 354–682 mAh g⁻¹ with low voltage hysteresis (Fig. 3D and Supplementary Fig. 7). As a contrast, the Li₂S@C cathode without MXene exhibits a low initial capacity of 498 mAh g⁻¹ due to poor $Li_2S$ utilization (Fig. 3C). Sluggish activation of this cathode takes as long as 10 cycles, followed by fast capacity decay to nearly 200 mAh g⁻¹ within 50 cycles, indicating poor redox activity and electrode reversibility.

The above results indicate the critical role of MXene in promoting the redox activity of bulk $Li_2S$ in Li₂S@MX cathode during cycling. XPS analysis of cycled Li₂S@MX cathode reveals the presence of Lewis acid-base interaction between $Ti_3C_2T_x$ MXene and LiPS by Ti-S signals at 455.6/461.4 eV and 161.2/162.3 eV in Ti 2p and S 2p spectra, respectively (Supplementary Fig. 8). The presence of sulfite (167.3 eV), thiosulfate (168.8 eV) and trace polythionates (170.4 eV) also suggest the LiPS interaction with oxygen-containing groups on MXene. Such chemical interactions effectively trap LiPS on the electrode interface, which not only inhibits LiPS loss to electrolyte but also promotes $Li_2S$ dissolution and electrode kinetics with LiPS as redox mediators[37–40]. The positive effect of MXene on $Li_2S$ dissociation and LiPS conversion is visualized by operando XRD analysis of working cells (Fig. 3E). For Li₂S@C, the diffraction from $Li_2S$ keep strong until over 95% depth of charge (DOC) while weak signals of sulfur appear in 80% DOC. These phenomena suggest the slow $Li_2S$ dissociation and LiPS conversion in this cathode. As a sharp contrast, high redox activity of Li₂S@MX allows the $Li_2S$ dissociation to be accomplished much soon in 60 % DOC to yield the sulfur in as early as 38% DOC. The interfacial charge transfer resistance ($R_{ct}$) and interface resistance ($R_{surf}$) of Li₂S@MX cathode remain stable with a low level in different DOC, as revealed by in situ electrochemical impedance spectroscopy (EIS) (Supplementary Fig. 9 and Supplementary Table 1). The improved redox kinetics and interfacial properties of bulk $Li_2S$ in Li₂S@MX cathode can be attributed to the MXene that well combines high chemical reactivity and conductivity (ca. 5600 S cm⁻¹)[41]. Their presence offers sufficient charge-accessible zones for strong adsorption of LiPS and in situ accomplishing their redox conversion without unnecessary diffusion to the conductive interface. This feature is distinct from common semiconductive or insulating LiPS absorbers without charge-accessible catalytic sites (e.g., metal oxides/sulfides, MOF, and polymers)[42–44]. Suppressed LiPS leaking from Li₂S@MX cathode is visualized by monitoring their evolution in species and concentration in working cells by operando UV-vis analysis (Fig. 3F). The LiPS with various chain lengths can be distinguished by the maxima of the first-order derivative of UV-vis spectra at different wavelengths[45,46]. Their concentrations are correlated with adsorption intensity following Beer-Lambert's law. The contour UV-vis pattern visualizes a much lower LiPS residue in Li₂S@MX-based working cells relative to that using Li₂S@C cathode. Upon charge, the Li₂S@MX

cathode releases the low-order LiPS thanks to the low activation potential barrier, followed by fast and sufficient conversion to long-chain ones with voltage rising, and vice versa during discharge. Whereas slow activation of Li₂S@C cathode releases rather low concertation of LiPS in the cells below 3.36 V. After that, the electrolyte contains mainly low-order LiPS due to sluggish conversion to long-chain ones. During discharge, continuous LiPS leaking results in a high residue of short-chain LiPS in the electrolyte even after cycling although the conversion of long-chain to short-chain LiPS proceeds fast. All these observations strongly evidence the important role of MXene in enhancing the redox activity, kinetics and reversibility of Li₂S@MX cathode.

## Fabrication and physicochemical characterizations of the composite gel polymer electrolyte

A quasi-solid-state CGPE with high ionic conductivity, mechanical robustness, thermal stability and wide electrochemical stability window is designed to strengthen the performance and reliability of anode-free cells. It is made of poly(vinylidene fluoride-co-hexafluoropropylene) (PVDF-HFP) containing LiTFSI and $Ti_3C_2T_x$ MXene with good conductivity, thermal conductance and mechanical strength. The CGPE has a porous texture with interconnected channels and uniform dispersion of MXene in the polymer matrix (Fig. 4A, B). Its thickness can be tailored to as thin as 20–30 μm without the sacrifice of flexibility and mechanical robustness (Supplementary Figs. 10, 11). The presence of MXene lowers the glass transition temperature ($T_g$) of MXene-free gel polymer electrolyte (GPE) by 5.3 °C, showing a reduction of polymer crystallinity for better chain dynamics and ionic transport (Supplementary Fig. 12). The ionic conductivity of such CGPE can be up to 0.81 mS cm⁻¹ by optimizing the ratio of LiTFSI (80%) and MXene (3.0 wt.%) in PVDF-HFP at 298.15 K (Supplementary Fig. 13A, B). It rises with temperature increasing according to the Arrhenius model (Supplementary Fig. 13C), suggesting the ionic transport in CGPE undergoes a rafting process decoupled from the long-range motion of polymer chains[47]. Adding 3% MXene into CGPE induces a slight rise of electronic conductivity to $9.3 \times 10^{-11}$ S cm⁻¹ from $1.0 \times 10^{-11}$ S cm⁻¹ for MXene-free GPE at $25 \pm 1$ °C (Supplementary Fig. 14). The MXene with a high elastic modulus of $330 \pm 30$ GPa also enhances the mechanical robustness and flexibility of CGPE, as characterized by over 103% strain elongation with high tensile strength in contrast to MXene-free GPE (19.8%) (Supplementary Fig. 15). These improvements allow the CGPE to effectively withstand the mechanical stain associated with Li dendrite growth and huge volume change of Li layer during repeated Li plating/stripping. Meanwhile, it also exhibits good thermal resistance and stability on flame by a synergy of PVDF-HFP with fluorinated side chains and MXene that can transform to fire-retardant $TiO_2$ on burning (Supplementary Movie 1–3). The CGPE can withstand a high temperature of 170 °C without shrinkage and cause no burning even on the flame at all (Fig. 4C, D). Upon heating, the thermal runaway on CGPE is rather even thanks to the homogenous dispersion of MXene with thermal conductance in it. Whereas the extensively used polypropylene (PP) separator shrinks at only 120 °C and immediately burns on fire, which certainly risks the cells in case of accidents (Fig. 4C, D). With a wide electrochemical stability window, the CGPE can withstand high voltage up to 4.7 V (vs. Li/Li⁺) (Supplementary Fig. 16), which makes it also compatible with commercial high-voltage cathodes such as $LiFePO_4$ and NMC811 to deliver specific energy of 595–793 Wh kg$_{cathode}^{-1}$ at specific power of 158–1398 kW kg$_{cathode}^{-1}$ (Supplementary Fig. 17).

Porous CGPE with high ionic conductivity and mechanical robustness can also effectively stabilize the Li metal anode without limiting the accessibility of electrode interface to homogenous Li⁺ flux. The symmetric Li||Li cells using CGPE (Li|CGPE|Li) exhibit no voltage polarization for 750 h at current densities of 1.0)2.0 mA cm⁻² and areal capacity of 2.0)4.0 mAh cm⁻² (Fig. 4E). Whereas the Li||Li cells using liquid electrolyte (Li|LE|Li) encounter rapid failure after 300 h by the short circuit under identical conditions (Supplementary Fig. 18). In Li||

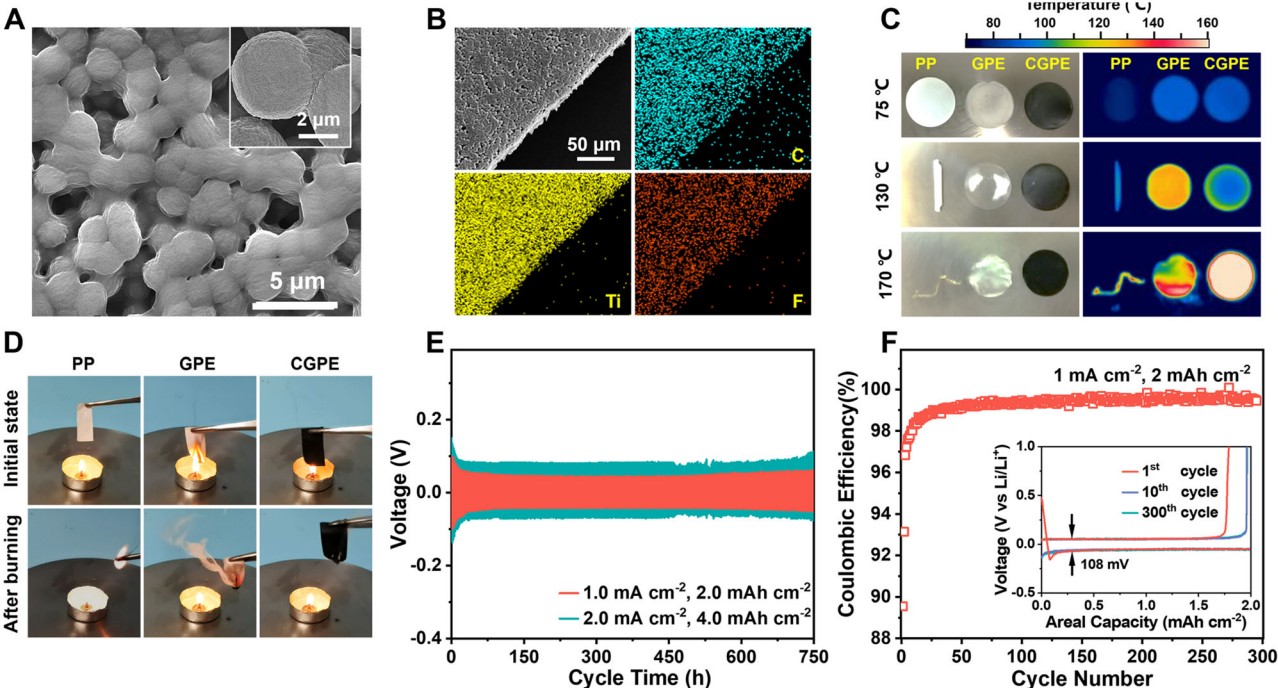

**Fig. 4 | Physicochemical and electrochemical characterizations of the CGPE.**
**A** SEM images of CGPE. **B** Elemental mapping of CGPE. **C** Thermal stability test of PP, MXene-free GPE and CGPE. The left is the optical photograph and the right is the infrared thermography. **D** Flame test of PP, MXene-free GPE and CGPE. **E** Voltage-time profiles of Li|CGPE | Li cells with a cycling capacity of 2.0 mAh cm$^{-2}$ at a current density of 1.0 mA cm$^{-2}$ and a cycling capacity of 4.0 mAh cm$^{-2}$ at 2.0 mA cm$^{-2}$. **F** Coulombic efficiency (CE) of Cu|CGPE | Li cell with a Li deposition capacity of 2.0 mAh cm$^{-2}$ at 1.0 mA cm$^{-2}$. The insert is the discharge-charge voltage curves at 1st, 10th and 300th cycles.

Cu cells using CGPE (Li|CGPE|Cu), the CGPE directs smooth Li plating on the lateral direction with a smooth mosaic surface on Cu by hindrance effect, in contrast to mossy Li growth in Li|LE|Cu cells (Supplementary Fig. 19)[48,49]. Such morphology is desired to reduce the formation of electrically isolated "dead Li", thereby enhancing the cell reversibility. Meanwhile, the CGPE with a higher Li$^+$ transport number ($t_{Li^+}$ = 0.57) than liquid electrolyte (0.47) can also facilitate uniform Li plating according to Sand's law (Supplementary Fig. 20)[50–52]. As a result, the Li|CGPE|Cu can exhibit a high CE of 99.2 % for 300 cycles with flat and long voltage plateaus and a small voltage hysteresis (Fig. 4F). This improvement is important to maintain high Li utilization in anode-free cells for achieving better reversibility and cycle life.

**Assembly and electrochemical energy storage performance of quasi-solid-state anode-free Li$_2$S|CGPE|Cu cells**
Quasi-solid-state anode-free Li$_2$S@MX|CGPE|Cu full cells are assembled by using Li$_2$S@MXene cathode against Cu current collector in CGPE. They exhibit an open-circuit voltage (OCV) at ca. 0.3 V with an operating voltage range of 1.7–2.8 V (Fig. 5A). The CVs of such cells show an anodic peak at ca. 2.54 V for Li$_2$S activation during the initial scan from the OCV to 3.5 V at a scan rate of 0.1 mV s$^{-1}$. The rest of the cycles feature with overlapping CVs with an anodic peak at ca. 2.4 V for Li$_2$S oxidation to sulfur and two cathodic peaks at ca. 2.3 and 2.0 V for sulfur reduction to LiPS and finally to Li$_2$S, suggesting a reversible LiPS-intermediated redox pathway likewise half cell (Supplementary Fig. 21). The discharge-charge curves of such quasi-solid-state cells also share a similar two-plateau feature with Li$_2$S@MXene cathode in half cells using liquid electrolytes. The Li$_2$S@MX|CGPE|Cu full cells limited by Li$_2$S cathode with mass loading of ca. 5.0 mg cm$^{-2}$ deliver a high initial charge and discharge capacity of 1023 and 819 mAh g$^{-1}$ at a specific current of 233.2 mA g$^{-1}$ with 80% CE, respectively (Fig. 5A). The initial Li loss is inevitable for the formation of solid electrolyte interphase (SEI), which is critical to guide subsequent smooth Li deposition

on the anode side and reduce its side reaction with the electrolyte[53,54]. This cell exhibits high capacity retention of 80% after 300 cycles at 233.2 mA g$^{-1}$. The charge-discharge voltage curves keep nearly constant with a narrow $\Delta E$ of 290 mV throughout cycling, reflecting the good reversibility with low polarization (Fig. 5A, B). When cycled at specific currents of 583 mA g$^{-1}$, it retains a capacity of 290 mAh g$^{-1}$ after 500 cycles, which represents an appealing performance compared to other anode-free cells assembled and tested using commercially available intercalation cathodes (Supplementary Figs. 17 and 22). In contrast, replacing the CGPE with liquid electrolyte in the same anode-free cell results in much lower capacity with rapid decay to 200 mAh g$^{-1}$ within 100 cycles due to LiPS shuttling and Li dendrite growth (Fig. 5B and Supplementary Fig. 23). Applying Li$_2$S@MX cathode with nearly doubled Li$_2$S loading (9.8 mg cm$^{-2}$) in Li$_2$S@MX|CGPE|Cu cells can deliver a high capacity of 688 mAh g$^{-1}$ (6.7 mAh cm$^{-2}$) with 84 % capacity retention for 180 cycles at a specific current of 233.2 mA g$^{-1}$ (Fig. 5B). Even tripling the Li$_2$S loading to as high as 14.6 mg cm$^{-2}$ can still reach a high initial capacity of 505 mAh g$^{-1}$ (7.32 mAh cm$^{-2}$) with stable capacity retention of 80 % for 180 cycles. The capacity rise during initial cycles is a result of the gradual activation of micrometer-sized Li$_2$S in thick electrodes in CGPE. Whereas the Li$_2$S@C cathode with similarly high Li$_2$S loading cannot charge at all due to poor redox activity with high electrical resistance. After electrochemical cycling, the CGPE well retains the original texture in terms of microstructure, composition and crystallinity without apparent TiS$_2$ formation, manifesting high electrochemical stability (Supplementary Figs. 24 and 25). It has been proposed that the CE in anode-free cells is firstly limited by the cathode with irreversible initial capacity loss, which leaves excess Li on the countered electrode (e.g., Cu). With continuously cycling, the irreversible Li loss will eventually take over to deteriorate the CE[8]. This undesired transition is delayed in quasi-solid-state anode-free Li$_2$S@MX|CGPE|Cu cells to secure high CE for over 300 cycles thank to good polysulfide retention and smooth Li plating (Supplementary

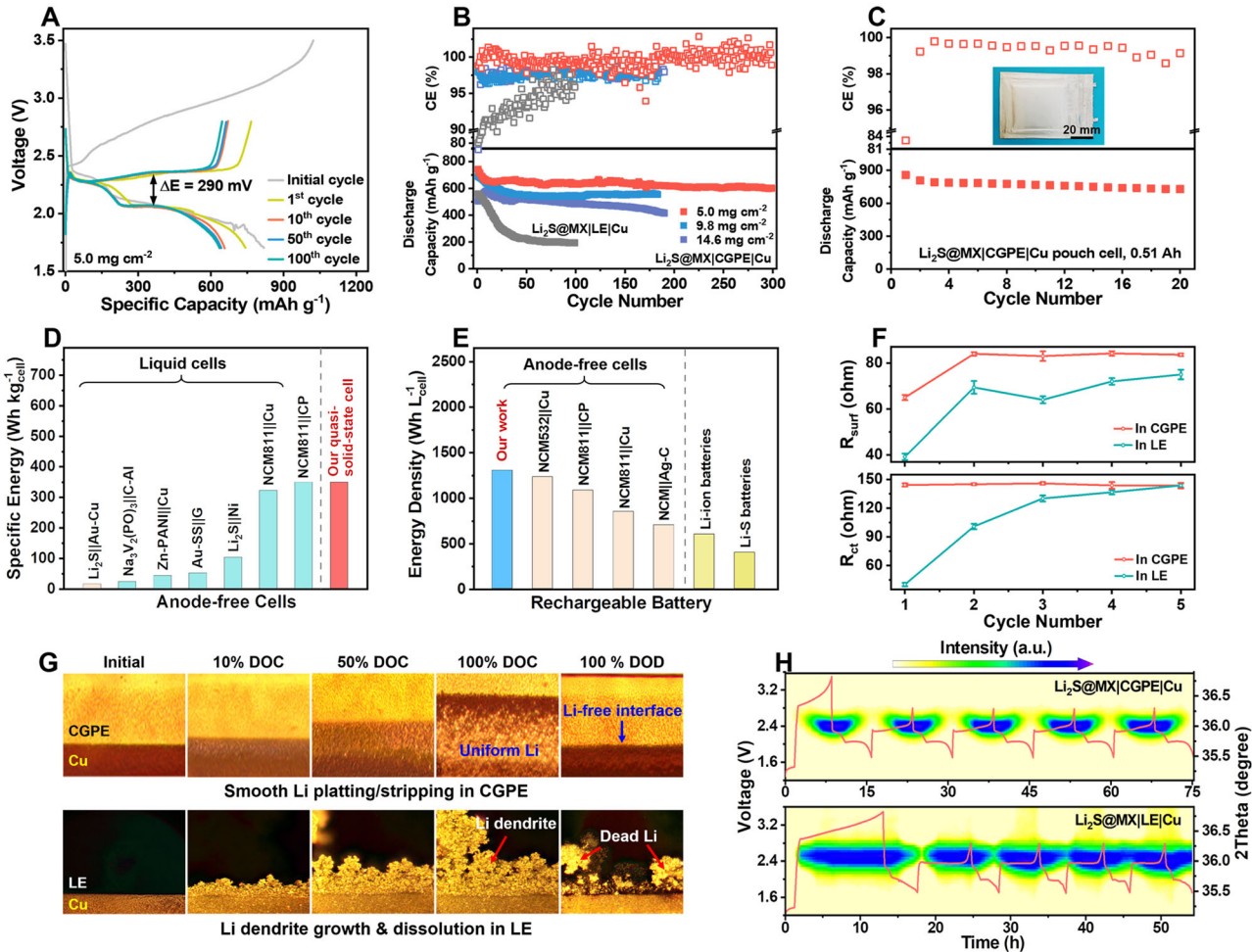

**Fig. 5 | Performance of quasi-solid-state anode-free Li2S@MX|CGPE|Cu cells.**
**A** Discharge-charge voltage curves of Li$_2$S@MX|CGPE|Cu cell at 1st, 10th, 50th and 100th cycles. **B** Cycling stability and CE of Li$_2$S@MX|CGPE|Cu and Li$_2$S@MX|LE|Cu cells at a specific current of 233.2 mA g$^{-1}$. **C** Cycling performance and CE of a 0.51 Ah pouch-type Li$_2$S@MX|CGPE|Cu cell at 233.2 mA g$^{-1}$. A comparison with reported anode-free cells and the state-of-the-art Li-ion and Li-S batteries in **D** specific energy and **E** energy density on cell level[55–63]. **F** In situ EIS revealing the evolution of R$_{ct}$ and R$_{surf}$ of Li$_2$S||Cu cells at different cycles in LE and CGPE. The error bars are based on three independent repeated experiments. **G** Operando optical images of Li deposition on the Cu foil surface at different states of charge/discharge in Li$_2$S@MX|CGPE|Cu and Li$_2$S@MX|LE|Cu full cells. The cells are initially charged from OCV to 3.5 V and then discharged from 3.5 to 1.7 V at a specific current of 233.2 mA g$^{-1}$ at 30 ± 1 °C. **H** Operando XRD contour patterns of Li$_2$S@MX|CGPE|Cu and Li$_2$S@MX|LE|Cu cells showing the Li evolution during initial five cycles and the corresponding voltage profiles. The cells are initially charged from OCV to 3.5 V and then cycled between 1.7 to 2.8 V at a specific current of 116.6 mA g$^{-1}$ at 30 ± 1 °C.

Fig. 26). The CGPE also works effectively in anode-free cells using Li$_2$S cathode with various mass loading (5.0–15.2 mg cm$^{-2}$) against Ni foil to deliver high capacities of 605 – 826 mAh g$^{-1}$ with capacity retention of 60–65% at a specific current of 233.2 mA g$^{-1}$ (Supplementary Fig. 27). The specific energy of Li$_2$S@MX|CGPE|Cu full cells is calculated to be 1423 Wh kg$_{cathode}^{-1}$ in terms of total active mass in the cathode (Supplementary Table 2). High specific energy of 364 Wh kg$_{cell}^{-1}$ can be achieved on cell level including the total mass of the cathode, Cu foil and CGPE. On this basis, the anode-free cell design also enables a calculated energy density of 1363 Wh L$_{cell}^{-1}$ based on the total volume of the above cell components.

Fresh pouch-type quasi-solid-state Li$_2$S@MX|CGPE|Cu cells exhibit negligible self-discharge for over 450 h due to the huge activation potential barrier of Li$_2$S, which extends the storage life of the battery (Supplementary Fig. 28). A high initial capacity of 808 mAh g$^{-1}$ can be achieved with a capacity retention of 60 % after 150 cycles at a specific current of 233.2 mA g$^{-1}$ (Supplementary Fig. 29). After 50 cycles, one of the pouch cells is disassembled at the fully charged state to analyze the morphology change of near-tab (R1), center (R2) and corner region (R3) on the anode (Supplementary Fig. 30A). The SEM observation reveals the formation of a dendrite-free Li layer with a relatively

smooth surface on all regions (Supplementary Fig. 30B–D). The Li-plated Cu foil punched from all these regions can still work effectively to deliver similar capacities of nearly 4 mAh cm$^{-2}$ against Cu current collectors in CGPE, indicating high effectiveness of CGPE on smoothing Li plating/stripping on large electrode area on cell level (Supplementary Fig. 30A). Larger 0.51 Ah pouch-type anode-free cells consisting of 4 layers of Cu foil sandwiched by 3 layers of Li$_2$S cathodes and CGPE could still retain high specific energy over 340 Wh kg$_{cell}^{-1}$ and energy density over 1323 Wh L$_{cell}^{-1}$ (Fig. 5C, Supplementary Fig. 31, Supplementary Tables 3 and 4). These battery performances are well-positioned in comparison with other anode-free cells reported in the literature and state-of-the-art Li-ion and Li-S batteries based on different chemistry (Fig. 5D, E)[55–63].

## Discussion
### Operando measurements and analyses of quasi-solid-state anode-free cells
High efficiency of CGPE for smoothing Li plating is validated by operando optical microscopy of Li$_2$S@MX|CGPE|Cu cells for the initial cycle. A reference cell is also assembled from Li$_2$S@MX cathode and Cu foil with liquid electrolyte as a comparison. Restricted by robust

CGPE, smooth Li plating/stripping occurs in Li$_2$S@MX|CGPE|Cu cells to maintain the interfacial properties stable during cycling, as reflected by stable $R_{ct}$ and $R_{surf}$ (Fig. 5F, Supplementary Fig. 32 and Supplementary Table 5). Upon charge, the thickness of Li layer on Cu substrate gradually increases, followed by complete Li stripping after discharge (Fig. 5G). The interface between Li and CGPE keeps flat without any deformation by Li dendrite growth throughout Li plating/stripping at a current density of 1.2 mA cm$^{-2}$. In contrast, the growth of granular Li is triggered on Cu foil in as early as 10 % DOC, which serves as the nucleation site to guide the mossy Li dendrite growth and irreversibly damage the electrode-electrolyte interface. After discharge, a high residue of "dead Li" is electrically isolated from the electrochemical system, causing low Li utilization, poor cell reversibility and rapid cell failure.

Operando XRD analysis is further performed to monitor the Li evolution on the Cu current collector in Li$_2$S@MX|CGPE|Cu full cell. The Li can be identified by the strong diffraction at 36.2 °, and the signal strength is correlated with its amount. Upon charge, the Li$^+$ released from Li$_2$S@MX cathode is deposited on Cu as Li metal in CGPE. Accordingly, the Li metal signals rise in the XRD pattern until the maximum intensity at the end of the charge (Fig. 5H). During the next discharge, the Li metal signals become weaker by Li stripping from Cu surface and completely vanish at a cutoff voltage of 1.7 V, showing full Li recovery back to the cathode. Symmetric and similar XRD patterns for continuous cycles visualize high reversibility of repeated Li plating/stripping in CGPE. On the contrary, the Li platting takes a longer time than stripping in liquid electrolytes, resulting in asymmetric and varied XRD patterns. The signals of Li metal exist even after discharge and become stronger with cycling. These phenomena reveal the poorly reversible Li plating/stripping in liquid electrolyte due to a continuous accumulation of electrically isolated "dead Li" on the anode (Fig. 5H). Moreover, the signal of LiH is also detected in a range of 37.8 ° in XRD pattern when cycling in the liquid electrolyte (Supplementary Fig. 33). A high accumulation of such species with electrochemical irreversibility, poor conductivity and much brittle structure than Li metal is harmful to the cyclability of anode-free cells[64]. For the cells using CGPE, however, the LiH is not detected on the anode due to the restricted side reactions of Li metal with electrolyte.

### Safety assessment of quasi-solid-state anode-free cells

Safety risk remains a significant obstacle towards the practical applications of the state-of-the-art LIBs and LMBs involving reactive Li metal anodes, radical oxygen-releasing cathodes and flammable liquid electrolytes[65]. The anode-free cell design may offer better reliability than LMBs due to strictly limited Li amount and the absence of reactive Li at a fully charged state. Applying robust and fireproof CGPE without leaking risk can further strengthen cell safety against abuse conditions. Fully charged soft-packaged pouch Li$_2$S@MX|CGPE|Cu cells exhibit negligible thermal runaway upon external short circuit and nail penetration, indicating high stability against electrical and mechanical abuse (Fig. 6A–C). In contrast, the Li-S cells undergo over 10 times higher temperature rise in two minutes after mechanical damage. After nail penetration or even cutting in air, the quasi-solid-state Li$_2$S@MX|CGPE|Cu cell could still work to power the LEDs without electrolyte leaking (Fig. 6D, Supplementary Movies 4, 5). Cutting a large part of fully charged cells in the air cause no violent reactions due to the protection of Li metal from the air by CGPE. The quasi-solid-state Li$_2$S@MX|CGPE|Cu cells also exhibit high stability upon overheating to 100 °C without apparent deformation thank to the use of thermal stable CGPE (Fig. 6E). Whereas Li-S cells with similar electrode loading encounter rapid swelling by the evaporation and decomposition of liquid electrolyte via vigorous side reactions with Li metal anode at high temperature. Applying the fireproof CGPE also allows the Li$_2$S@MX|

CGPE|Cu cells to maintain stable energy output on the flame without burning in contrast to the immediate and violent combustion of Li-S cells on fire (Fig. 6F, Supplementary Movies 6, 7). With flexible design enabled by CGPE, the Li$_2$S@MX|CGPE|Cu pouch cells can maintain stable energy output under repeated folding and rolling, holding promise in high-energy flexible and wearable devices (Fig. 6G).

In summary, we demonstrated a quasi-solid-state anode-free cell assembled from high-capacity Li$_2$S cathode and robust composite gel electrolyte for meeting high energy and safety. A cold pressing strategy is developed to activate bulk Li$_2$S within MXene matrix full of polar groups, which ensures high redox activity to minimize the charge difficulty of high loading Li$_2$S@MX cathode. The MXene contributes to not only rapid Li$_2$S dissociation and conversion but also good mechanical robustness, ionic conductance, thermal stability and fire retardancy of gel polymer. Such gel electrolyte works effectively in directing smooth Li plating/stripping with dendrite growth while inhibiting LiPS shuttling. Applying Li$_2$S@MX cathode into such electrolyte creates a quasi-solid-state anode-free cell with good reversibility, high energy density and long life. Moreover, this cell also exhibits low self-discharge and safety during abuse tests by conducting stable redox chemistry without reactive oxygen and excess Li inventory in robust fireproof gel electrolyte.

## Methods

### Synthesis of Ti$_3$C$_2$T$_x$ MXene

The Ti$_3$C$_2$T$_x$ MXene was synthesized by etching 5.0 g of Ti$_3$AlC$_2$ MAX phase (400 mesh, Yuehuan Technology Co. Ltd., Shanghai, China) in HCl solution (6 M, 200 mL) dissolved with 13.2 g LiF (Sigma Aldrich, 99.98%) for 48 h at 40 °C. After several centrifugation-rinsing cycles with deionized (DI) water, the products were dispersed in 150 mL DI water and kept under ultrasonic for 3 h. The dark green supernatant was collected by centrifuging at 2000 rpm for 1 h and was dispersed in DI water, followed by freeze-drying.

### Synthesis of Li$_2$S@MX cathode

Commercial micrometer-sized Li$_2$S powder (Sigma Aldrich, 99.98%) and Ti$_3$C$_2$T$_x$ MXene were finely mixed with a weight ratio of 7: 3 by ball milling in the zirconia pot with twelve zirconia balls (3 mm in diameter) in Ar atmosphere. The above mixture was pressed under a constant pressure of 300 MPa using a customized chrome steel mold with a cold pressing machine to form free-standing Li$_2$S@MX cathodes. Varying the amount of mixture powder for cold press can readily tune the areal mass loading of the cathode. For example, the amounts of Li$_2$S and MXene mixture are set as 11.0–32.1 mg to reach Li$_2$S loading of 5.0–14.6 mg cm$^{-2}$ for making the cathode with an average thickness of 40–115.8 µm, respectively. As a comparison, a Li$_2$S@C cathode was also made in a similar way except to replace the MXene by carbon black (particle size: 20–60 nm, >99%, Kejing Star Co., Ltd., Shenzhen, China).

### Synthesis of composite gel polymer electrolyte (CGPE)

The poly(vinylidene fluoride-co-hexafluoropropylene)(PVDF-HFP, $M_w$ = 455,000, Aladdin Biochemical Co., Ltd., Shanghai, China, 400.0 mg) was dissolved in N-methyl-2-pyrrolidone (NMP, > 99.9%, Canrd New Energy Co., Ltd., Guangdong, China, 3.0 mL) under vigorous stirring at 75 °C, followed by adding a NMP colloid of Ti$_3$C$_2$T$_x$ MXene (10 mg mL$^{-1}$, 2.0 mL) and LiTFSI (>99.8%, Dodo Chem Co., Ltd., Suzhou, China, 320 mg). The resultant suspension was cast onto a customized polytetrafluoroethylene (PTFE, > 99.8%, Dodo Chem Co., Ltd., Suzhou, China) plate to obtain CGPE after removal of the solvent by evaporation at 75 °C in vacuum for 24 h. For compassion, the gel polymer electrolyte without MXene was also prepared in a similar way in the absence of MXene.

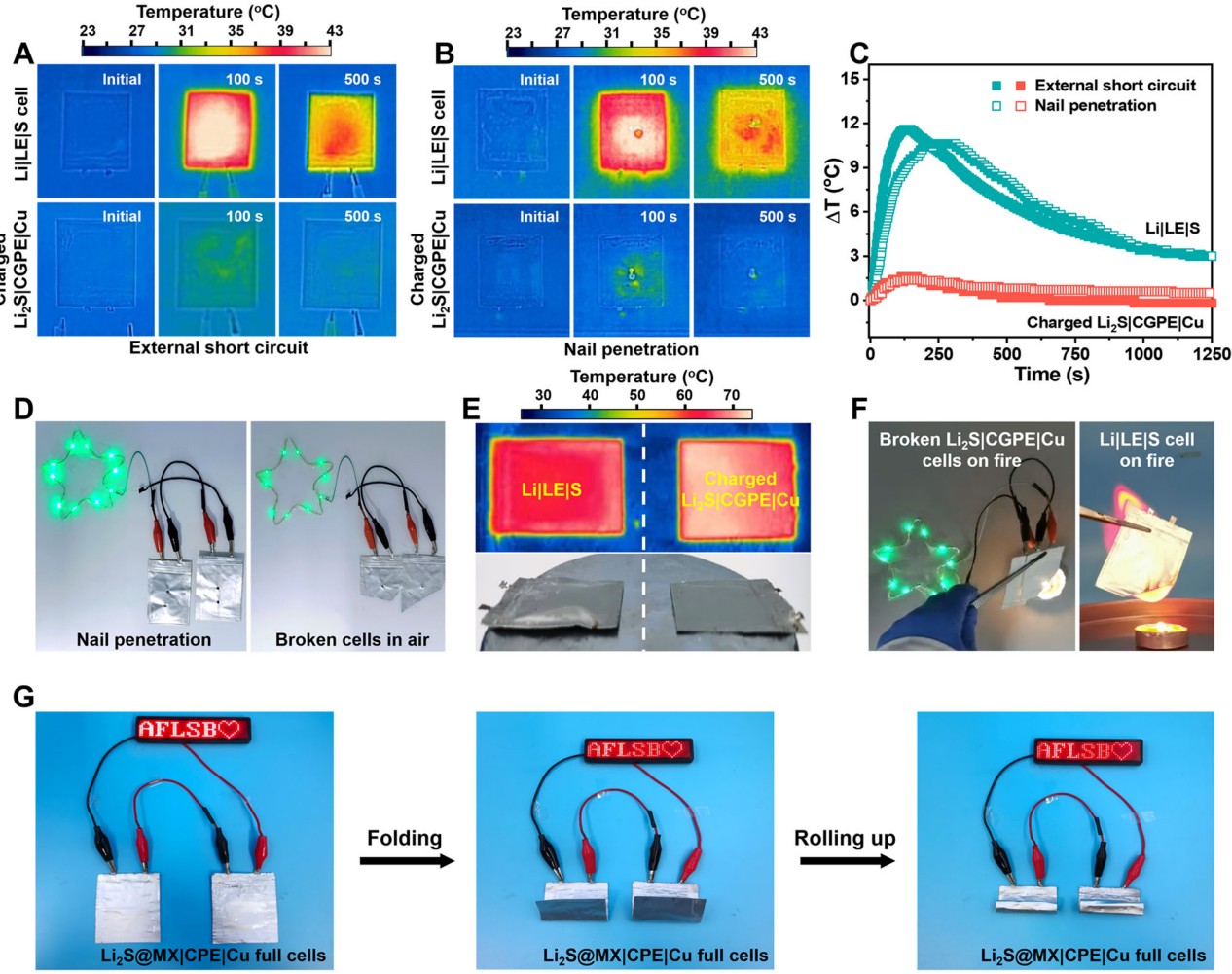

**Fig. 6 | Mechanical and thermal abuse tests of quasi-solid-state Li2S@MX|CGPE| Cu pouch cells.** Infrared thermography of charged Li2S@MX|CGPE|Cu full cells and Li|LE|S cells after **A** external short circuit and **B** nail penetration. **C** The temperature vs. time curves of charged Li2S@MX|CGPE|Cu full cells and Li|LE|S cells after external short circuit and nail penetration. **D** Optical images of the LEDs powered by Li2S@MX|CGPE|Cu full cells after nail penetration and the next cutting in air. **E** Infrared thermography and optical image of charged Li2S@MX|CGPE|Cu and Li| LE|S cells under overheating conditions. **F** The flame test of charged Li2S@MX| CGPE|Cu and Li|LE|S cells. **G** Optical images of the LED lighted by two series-connected Li2S@MX|CGPE|Cu cells under rolling or folding conditions.

## Assembly and testing of Li metal cells

CR2032 coin cells were used for the tests. All the Li metal cells measurements are carried out on a LAND CT2001A battery tester by using Li foil as the counter and reference electrode. The free-standing Li2S@MX was directly used as the working electrode. A solution of 1.0 M lithium bistrifluoromethanesulfonylimide (LiTFSI) in 1, 3-dioxolane (DOL) and 1, 2-dimethoxyethane (DME) (1:1 by volume) with 2.0 wt.% LiNO$_3$ additive was used as the liquid electrolyte (<20 ppm H$_2$O). The ratio between the electrolyte and cathode was fixed to 4 μL mg$_{Li2S}$$^{-1}$. Polypropylene-polyethylene (PP) separator with a thickness of 20 μm and 41.5 % porosity (Canrd New Energy Co., Ltd., Guangdong, China) was used for the cells using liquid electrolyte. For comparison, the Li metal cells were also assembled under identical conditions by using CGPE with an average thickness of 20 μm. The Li metal cells were firstly charged to 3.5 V at a specific current of 116.6 mA g$^{-1}$ to activate the Li2S@MX cathode on a LAND CT2001A battery tester. Afterward, the galvanostatic discharge-charge tests were carried out between 1.7–2.8 V at various specific currents, where all the specific capacities were calculated based on Li2S mass. The CV tests were performed from 1.7 to 3.5 V for the initial scan and between 1.7–2.8 V for the rest cycles at a scan rate of 0.1 mV s$^{-1}$ using a Vertex.C.EIS electrochemical workstation (IVIUM). All the cells were tested at 30 ± 1 °C.

The commercial LiFeO$_4$ (LFP, particle size: 1.3 ± 0.5 μm, >99%), NMC811 (particle size: 4.0 ± 1.0 μm, >99%), carbon black (Super P) and polyvinylidene difluoride (PVDF) were purchased from KeJing Star Co., Ltd., Shenzhen, China. The Li metal cells were also assembled by using LFP or NMC811 cathodes. These cathodes were made by casting the slurry mixture of the active material (LFP or NMC811), Super P and PVDF in a weight ratio of 8:1:1 in NMP on Al foil, followed by drying at 80 °C in vacuum for 24 h. The mass loading of LFP and NMC811 cathode is ca. 26 and 22 mg cm$^{-2}$, respectively. The average thickness of the LFP and NCM811 cathodes are 250 and 240 μm, respectively. A trace amount of liquid electrolyte (0.3 μL mg$_{active\ material}$$^{-1}$) was added between the cathode and CGPE with an average thickness of 20 μm to improve the interfacial compatibility. Galvanostatic discharge-charge tests were performed between 2.5–3.8 V for LFP cathode and 3.0–4.3 V for NMC811 cathode at various specific currents at 30 ± 1 °C.

## Assembly and tests of asymmetric Cu‖Li and symmetric Li‖ Li cells

The Cu‖Li cells and symmetric Li‖Li cells were assembled by using Li foil with a thickness of 400 μm (>99.95%, Medium Energy Lithium Co., Ltd., Tianjin, China) or Cu foil with a thickness of 4.5 μm (> 99.98%, KeJing Star Co., Ltd., Shenzhen, China) as the working electrode against Li foil as the counter and reference electrode in CGPE with an

average thickness of 20 μm, respectively. The Cu‖Li cells were firstly galvanostatically discharged to a certain capacity for Li plating and then charged to 1 V for Li stripping. The symmetric Li‖Li cells were cycled with controlled capacities and current densities. All the cells were tested at $30 \pm 1\,°C$.

## Assembly and tests of anode-free cells

The anode-free coin cells were assembled by using $Li_2S@MX$ with various mass loading as the cathode against Cu foil in CGPE. A trace amount of liquid electrolyte (LiTFSI in DOL/DME, 1.5 μL $mg_{Li2S}^{-1}$) was applied to improve the interfacial compatibility between the cathode and CGPE. The cells were firstly charged to 3.5 V at a specific current of 116.6 mA $g^{-1}$ to activate the $Li_2S@MX$ cathode on a LAND CT2001A battery tester. Afterward, the galvanostatic discharge-charge tests were carried out between 1.7–2.8 V at 233.2 mA $g^{-1}$. All the specific capacities were calculated based on $Li_2S$ mass.

The quasi-solid-state anode-free coin cells were also assembled by using the above LFP or NMC811 cathodes against Cu foil in CGPE with an average thickness of 20 μm. The mass loading of LFP and NMC811 cathode is ca. 26 and 22 mg $cm^{-2}$, respectively. The average thickness of the LFP and NCM811 cathodes are 250 and 240 μm, respectively. The active mass loading of NCM811 for anode-free cells was 22.0 mg $cm^{-2}$ with an average thickness of 240 μm. A trace amount of liquid electrolyte (0.3 μL $mg_{active material}^{-1}$) was added between the cathode and CGPE to improve the interfacial compatibility. As a comparison, the anode-free cells were also assembled by a similar way except to use liquid electrolyte to replace CGPE. The liquid electrolyte is 1 M LiTFSI in DOL/DME with 2.0 wt.% $LINO_3$ or 1 M $LiPF_6$ in EC/DEC with better stability at high voltage for the cells using LFP or NMC811 cathode, respectively. Galvanostatic discharge-charge tests were performed between 2.5–3.8 V for LFP cathode and 3.0–4.3 V for NMC811 cathode at various specific currents at $30 \pm 1\,°C$.

The 0.51 Ah $Li_2S@MX \,|\, CGPE \,|\, Cu$ pouch cells were assembled by alternately stacking 4 layers of Cu sheets, 3 layers of double-side $Li_2S@MX$ cathodes (4.8 mg $cm^{-2}$) and 6 layers of CGPE. The cells were sealed with Al plastic in an Ar-filled glove box ($H_2O < 0.1$ ppm, $O_2 < 0.1$ ppm). The pouch cells were firstly charged to 3.5 V at a specific current of 116.6 mA $g^{-1}$ to activate the $Li_2S@MX$ cathode. Afterward, the galvanostatic discharge-charge tests were carried out between 1.7 – 2.8 V at 233.2 mA $g^{-1}$ at $30 \pm 1\,°C$ under a pressure of 170 kPa.

## Measurement of Li-ion transport numbers ($t_{Li^{(+)}}$) and electronic conductivity ($\sigma_{e^-}$) of CGPE

The $t_{Li^{(+)}}$ of CGPE were calculated by direct-current (DC) polarization of Li|CGPE|Li symmetric coin cells by the Bruce-Vincent-Evans equation:

$$t_{Li^{(+)}}(=) \frac{I_S}{I_0} \times \frac{(\triangle V - I_0 \times R_0)}{(\triangle V - I_s \times R_s)} \qquad (1)$$

where $\Delta$V is the applied DC voltage pulse (10 mV), $I_0$ and $I_S$ are the initial and steady-state currents, and $R_0$ and $R_S$ are the corresponding initial and steady-state resistances obtained by alternating current impendence, respectively. The $t_{Li^{(+)}}$ of LE was calculated from Li|LE|Li symmetric cells by a similar way. All these tests were performed on a Vertex.C.EIS electrochemical workstation (IVIUM).

## Materials characterization

The morphology of the materials was investigated by transmission electron microscopy (TEM, FEI Tecnai G20) and scanning electron microscopy (SEM, JEOL JSM-7800F) with an energy dispersive spectrometer. X-ray diffraction (XRD, Rigaku D/MAX-2400, Cu Kα), XPS (Thermo ESCALAB MK II) and Raman (Horiba LabRAM) spectra were employed for analysis of the texture and composition of the samples. For ex situ characterizations, the electrodes or CGPE were

disassembled from the cycled cells, followed by repeated rinsing with DME and drying at 60 °C overnight in an Ar-filled glovebox. Afterwards, they were sealed into a vacuum transfer box for transport to the equipment. The infrared thermography images of the cells were captured by a thermal infrared camera (FLIR, C5). The mechanical properties of the CGPE were measured by a compression tester (Instron 5567).

## Operando measurements and analyses

The operando analysis of Li metal cells was performed by using $Li_2S@MX$ as the working electrode against Li foil in the liquid electrolyte (LiTFSI in DOL/DME with 2.0 wt.% $LiNO_3$). For anode-free cells, the operando analysis was conducted with $Li_2S@MX$ as the working electrode against Cu foil in CGPE. The operando UV-vis analysis was conducted by using a customized cell with a quartz window on UV-vis spectrometer (PerkinElmer Lambda 750). The UV-vis spectra were recorded per 600 s in a wavelength range of 350–700 nm during cell cycling between 1.7–2.8 V at a specific current of 116.6 mA $g^{-1}$. The operando X-ray diffraction patterns were collected by using a home-made cell with a Be window for per 360 s in a 2θ range of 5 to 70 ° for half cells and per 600 s in a 2θ range of 20 to 50 ° for full cells on X-ray diffractometer (Bruker D8 DISCOVER) with a 2D detector. Meanwhile, the batteries were cycled between 1.7–3.5 V at 116.6 mA $g^{-1}$. In situ EIS measurements were measured on a Vertex.C.EIS electrochemical workstation (IVIUM). The charging process was suspended every 60 min, and the cells were rested for 60 min to minimize the polarization before EIS measurement. The EIS spectra were recorded in a frequency range of 100 kHz to 10 MHz with an amplitude of 5 mV (6 data points per decade) at different voltages. The operando optical observation of Li deposition behavior in anode-free cells was conducted by metallographic microscopy (NMM-800RF, China) using homemade cells with a quartz window.

## Computational method

The Vienna Ab Initio Package (VASP) was employed to perform all the density functional theory (DFT) calculations within the generalized gradient approximation (GGA) using the PBE formulation. The projected augmented wave (PAW) potentials were chosen to describe the ionic cores and take valence electrons into account using a plane-wave basis set with a kinetic energy cutoff of 400 eV. Partial occupancies of the Kohn−Sham orbitals were allowed using the Gaussian smearing method and a width of 0.05 eV. The electronic energy was considered self-consistent when the energy change was smaller than $10^{-5}$ eV. Geometry optimization was considered convergent when the force change was smaller than 0.02 eV/Å. Grimme's semiempirical DFT-D3 methodology was employed to describe the dispersion interactions between the $Li_2S_n$ ($n = 2, 4$ and 6) clusters and material surfaces.

The equilibrium lattice constants of hexagonal $Ti_3C_2$ monolayer in a vacuum layer of 20 Å in total were optimized to be $a = b = 3.076$ Å when using a $13 \times 13 \times 1$ Monkhorst-Pack $k$-point grid for Brillouin zone sampling. It was then used to build $4 \times 4$ supercells in the $x$ and $y$ directions. Two monolayers of O atoms were added to cover the two outmost Ti atomic layers for building the $Ti_3C_2O_2$ model. During structural optimizations, the $\Gamma$ point in the Brillouin zone was used for $k$-point sampling, and all atoms were allowed to relax. The nudged elastic band (NEB) method was employed to determine the kinetic barriers and transition state of the elementary reaction step $Li_2S \rightarrow LiS + Li$. The adsorption energy ($E_{ads}$) of adsorbate A can be calculated as $E_{ads} = E_{A/surf} - E_{surf} - E_{A(g)}$, where $E_{A/surf}$, $E_{surf}$ and $E_{A(g)}$ are the total energy of adsorbate A adsorbed on the surface, the energy of the clean surface, and the energy of isolated A molecule in a cubic periodic box with a side length of 20 Å and a $1 \times 1 \times 1$ Monkhorst-Pack k-point grid for Brillouin zone sampling, respectively.

## Reporting summary

Further information on research design is available in the Nature Research Reporting Summary linked to this article.

## Data availability

The data generated in this study have been provided in the manuscript or its Supplementary Information. Source data are provided with this paper.

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

## Acknowledgements

Z.Y.W. is grateful to the support of the National Natural Science Foundation of China (NSFC, No. 51972040, 51772040), Valiant Co. Ltd., Open project of State Key Laboratory of Organic-Inorganic Composites (oic-202201003), Talent Program of Rejuvenation of the Liaoning (No. XLYC1807032) and Innovation Support Program of Dalian City (No. 2018RJ04) for this work.

## Author contributions

Z.Y.W. and Y.Z.L. conceived the idea and co-wrote the manuscript. Y.Z.L. and X.Y.M. performed the experiments and theoretical calculations. Z.Y.W. and J.S.Q. guided all aspects of the work.

## Competing interests

The authors declare no competing interests.
