## [Peer Review File · Nature Communications]

REVIEWERS COMMENTS

Reviewer #1 (Remarks to the Author):

The article entitled Fireproof Quasi-solid-state Anode-free Lithium Battery with High Volumetric Energy by Liu et al. describes a Cu-Li₂S cell by using a MXene doped fluorinated polymer. The results are interesting but preliminary for publication. A major revision is necessary before to consider the manuscript for publication in Nature Comm.

Here my comments:

1) As first the author should test other cathodes: a) LiFePO₄ and b) a high energy density cathode as NMC 811 and add electrochemical results (high loading > 2 mAh/cm²). They should perform Ragone test and stability cycling (C/2-C/2 and 1C-1C). The tests are very important to characterize the GPE. When they use Li₂S as active material the stability curves are made at 0.2 C (Figure 5): the authors should increase the C rate (1C-1C) and shows the results over 500 cycles.

2) The role of MXene is unclear: the authors should add SEM image, EDS mappings and XRD patterns of GPE before and after cycling. Are there any side reactions occurred to MXene due to reactions with polysulfides? Is TiS₂ formed? By using other fillers, do the authors observe the same results? The authors should cite some previous reports about polysulfides retention: a) ACS Appl. Mater. Interfaces 12 (2020), 43560–43567

b) The Journal of Physical Chemistry C 122 (2), 2018, 1014-1023; c) Journal of Power Sources 427 (2019), 201-206; d) Energy Fuels 2020, 34, 8, 10188–10195; e) Chemical Engineering Journal 330 (2017) 644–650;

3) Why do plating is more uniform with GPE with respect to LE? Is it related to Li⁺ ion transport number? Some review articles should be cited: a) Angew. Chem. Int.Ed. 55, 2016, 500–513; b) Materials Science and Engineering: R: Reports 134, 2018, 1-21; c) Journal of Energy Chemistry 56 (2021) 420–437;

4) What is the role of 2% LiNO₃? When Li metal is used, LiNO₃ oxidize Li forming LiNOX. What happens when Cu metal is used? The authors should characterize by XPS the Cu foil after lithium metal plating and define a mechanism.

5) As comparison the authors should test nickel metal foil instead of copper by using same different mass loading as shown in Figure 5b.

6) As well the authors should report half cell in manuscript. In Figure: S5 and S6 what is the loading used? The authors should test cathode with different loadings. Moreover, the final results should be

compared with data cycling in Figure 5: it seems that the cells with a Cu anode show a better coulombic efficiency than half cells: why? In theory the results should be the opposite.

7) What is the mechanism of polysulfide retention? The authors should add a schematic. Probably the active species are Li_2S_6 and Li_2S_4 . The following reports should be used as reference a) *Nanoscale*, 2016, 8, 13638-13645; b) *Sustainable Energy & Fuels*, 2017, 1 (4), 737-747; c) *Journal of Energy Chemistry* 54, 2021, 434-451; d) *Journal of Power Sources*, 2016, 319, 247-254;

Reviewer #2 (Remarks to the Author):

This manuscript demonstrated an anode-free gel-electrolyte Li-S batteries using a $\text{Li}_2\text{S}/\text{MXene}$ composite cathode. The key points of the study including (1) MXene is a superior conductive additive (than carbon) for Li_2S cathode and good catalysts for Li_2S oxidation and the sulfur reduction and (2) semi-solid electrolyte made of PVDF-HFP incorporated with LiTFSI, LiTFSI-based DOL/DME electrolyte, and small amount MXene can enable efficient Li deposition-stripping thus bring anode free. However, the mechanisms are not new, MXene catalysts for sulfur cathode, gel polymer electrolyte, and anode-free concept have all been reported before. Therefore this manuscript should be submitted to a different journal with more specific scope.

Furthermore, some results in the current manuscript are vague:

(1) Figure 2F displays the calculated energy profile for Li_2S dissociation, but it is unclear why the reaction is Li_2S to $\text{LiS} + \text{Li}$: LiS does not exist.

(2) Mixing MXene into the polymer electrolyte (even with a low content) would increase the electronic conductivity of the electrolyte, which is not ideal.

(3) The combination of the Li_2S loading and the Li_2S specific capacity is low in the full cells. It is unlikely the full cell can deliver a specific energy of 364 Wh/kg. It is very vague how this value was calculated. Same goes with the calculated 1363 Wh/L energy density. The authors must be very specific and detailed on the calculation of these numbers.

(4) Similarly, the size of the full cells in this work seems very small, therefore the comparisons of specific energy and energy density in Figure 5c and 5d are incorrect. Because at least some of the values from the cited studies are from high capacity practical pouch cells (such as Ref 45).

Reviewer #3 (Remarks to the Author):

The surging development of electric vehicles and portable electronic devices eagerly desires powerful but thinner batteries that can store more energy in limited space with high reliability. The quasi-solid-state anode-free batteries developed in this work perfectly meet this desire due to impressive volumetric energy and safety against abuse. Stable redox of ultrahigh-loading Li₂S cathode in fireproof gel polymer electrolyte offers new opportunities of harvesting both excellent volumetric energy and safety. Direct use of commercial bulk Li₂S without tedious material fabrication further adds the practical value. The strategy applying robust polymer electrolyte to zero-excess Li metal also inspires one to address the Li dendrite issues in Li metal batteries. Such an innovative cell design holds promise in not only developing thinner cells but also breaking the safety bottleneck threatening the practical use of alkali-metal based batteries. I encourage the publication of this work in Nature Communications with a necessary revision of several points.

1. More evidence such as CVs is necessary to clarify the reaction pathway of anode-free Li₂S | Cu full cells.
2. It has been reported that the Li₂S cathode can be paired with high-capacity anodes such as Si for manufacturing high-energy batteries. Can the author comment on the potential of this kind of battery?
3. What is the interaction between LiPS and MXene? Please specify to avoid any misunderstanding.
4. In Figure 5B, the discharge capacity of full cells with higher Li₂S loading increases in the initial cycles. What is the reason?
5. Please provide the ionic conductivity of CGPE at different temperature.
6. The label of the y-axis in Figure 2F may be "reaction pathway". It would be better to just present the reaction equation here for better readability.

We highly appreciate the editor and the reviewers for their time and very valuable comments in improving the quality of this manuscript. Provided below is our detailed response to every question.

Reviewer # 1

The article entitled Fireproof Quasi-solid-state Anode-free Lithium Battery with High Volumetric Energy by Liu et al. describes a Cu-Li₂S cell by using a MXene doped fluorinated polymer. The results are interesting but preliminary for publication. A major revision is necessary before to consider the manuscript for publication in Nature Comm.

Our Response: We highly appreciate the reviewer for the encouraging comments and very constructive suggestions.

Q1. As first the author should test other cathodes: a) LiFePO₄ and b) a high energy density cathode as NMC811 and add electrochemical results (high loading > 2 mAh/cm²). They should perform Ragone test and stability cycling (C/2-C/2 and 1C-1C). The tests are very important to characterize the GPE. When they use Li₂S as active material the stability curves are made at 0.2 C (Figure 5): the authors should increase the C rate (1C-1C) and shows the results over 500 cycles.

Our Response: We thank the reviewer for the valuable suggestion. We firstly verified the wide electrochemical stability window of CGPE by linear sweep voltammetry (LSV). The CGPE can maintain high stability at high voltage up to 4.7 V (vs. Li/Li⁺). This feature makes it compatible with commercial high-voltage cathodes such as LiFePO₄ (LFP) and NMC811. Related data have been added as Supplementary Figure 16 in the revised Supplementary Information with necessary discussion in the revised MS, page12, highlighted in yellow.

Figure 1. LSV of CGPE at a scan rate of 0.2 mV s⁻¹ from 0 to 5 V.

We then evaluated the performance of LFP and NMC811 cathodes (> 3 mAh cm⁻²) in CGPE at 0.5 – 1.0 C (1 C = 170 mA g⁻¹ for LFP or 275 mA g⁻¹ for NMC811). Both cathodes exhibit stable capacity retention in CGPE in half cells with excess Li. No performance decay caused by CGPE is identified, validating the good electrochemical stability of CGPE against high-voltage cathodes again. In anode-free cells using these cathodes, utilization of CGPE also effectively improves the Coulombic efficiency

and lifetime in contrast to fast failure of anode-free cells using liquid electrolyte (LE) by uncontrolled Li dendrite growth and side reactions. When cycled at various current rates of 0.2 – 5.0 C, anode-free LFP||CGPE||Cu cells exhibit good capacity retention with high energy of 595 – 793 Wh kg_{cathode}⁻¹ at specific power of 158 – 1398 kW kg_{cathode}⁻¹. These results highlight the high compatibility of CGPE with commercial cathodes for developing anode-free cells although there is still a large room for performance improvement. Related data have been added as Supplementary Figure 17 in the revised Supplementary Information with necessary discussion in the revised MS, page12, highlighted in yellow.

Figure 2. (A) Cycling performance of anode-free LFP||CGPE||Cu or LFP||LE||Cu cells and LFP||CGPE||Li half cell between 2.5 – 3.8 V at 0.5 – 1 C (1 C = 170 mAh g⁻¹); (B) Cycling performance of anode-free NMC811||CGPE||Cu or NMC811||LE||Cu cells and NMC811||CGPE||Li half cell between 3.0 – 4.3 V at 0.5 C (1 C = 275 mAh g⁻¹). (C) Rate capability and (D) Ragone plot of anode-free LFP||CGPE||Cu cell at various current rates. The specific energy and power are calculated in terms of the active mass in the cathode.

One of the advantages of Li₂S cathode is the far superior theoretical capacity (1166 mAh g⁻¹) to commercial cathodes such as LFP (170 mAh g⁻¹) and NMC811 (275 mAh g⁻¹). It allows the Li₂S cathode with the same mass loading to induce > 4.2 – 6.8 times higher current densities than LFP and NMC811 cathodes for reaching the same C rate. A 0.2 C rate for cycling Li₂S cathode in our original MS induces a current density of 233.2 mA g⁻¹, corresponding to 1.36 and 0.84 C for LFP and NMC811 cathodes, respectively. Therefore, the C rate employed in our work is reasonable to evaluate the performance of anode-free cells.

Long-term cycling stability of anode-free cells is extremely challenging due to zero Li excess but inevitable Li loss in it. As a result, a lifetime of 500 cycles has been rarely achieved by such cells. Nevertheless, we tested the anode-free Li₂S@MX||CGPE||Cu cells at a higher current rate of 0.5 C (583 mA g⁻¹) according to the reviewer's valuable suggestion. After 500 cycles, it can still deliver a

far higher capacity (290 mAh g^{-1}) beyond even the theoretical capacities of LFP and NMC811 cathodes, showing great promise in delivering high energy. When cycled at 1.0 C, short-circuit occurs due to the intrinsic limitation of commercial Cu foil with poor lithophilicity. The work is undergoing to address this problem by surface modification of Cu or lithiophilic current collector in terms of literature reports (e.g., *Nat. Commun.* 2019, 10, 1896; *Adv. Energy Mater.* 2022, 220030; *Nat. Nanotechnol.* 2019, 14, 594). Related data have been added as Supplementary Figure 22 in the revised Supplementary Information with necessary discussion in the revised MS, page 14, highlighted in yellow.

Figure 3. Cycling performance of $\text{Li}_2\text{S}||\text{CGPE}||\text{Cu}$ cell at 0.5 C (583 mA g^{-1}).

Q2. The role of MXene is unclear: the authors should add SEM image, EDS mappings and XRD patterns of GPE before and after cycling. Are there any side reactions occurred to MXene due to reactions with polysulfides? Is TiS_2 formed? By using other fillers, do the authors observe the same results? The authors should cite some previous reports about polysulfides retention: a); b) *The Journal of Physical Chemistry C* 122 (2), 2018, 1014-1023; c) *Journal of Power Sources* 427 (2019), 201-206; d) *Energy Fuels* 2020, 34, 8, 10188–10195; e) *Chemical Engineering Journal* 330 (2017) 644–650;

Our Response: We appreciate the reviewer for the constructive suggestion. The MXene plays multiple roles in improving the properties of CGPE: i) enhance its ionic conductivity by reducing the crystallinity but improving chain migration of polymer; ii) strengthen it by high elastic modulus ($330 \pm 30 \text{ GPa}$); iii) improve thermal distribution within it to reduce thermal-induced shrinkage; v) convert to inflammable TiO_2 on the flame to enable its fire retardancy.

SEM and EDS data of CGPE before cycling have been available in Figure 4A and Figure 4B in the original MS. On this basis, the CGPE before and after cycling were further analyzed by XRD, XPS, SEM and EDS. It reveals that the CGPE well retains the original texture in terms of microstructure, composition and crystallinity without apparent side reactions. Both XPS and XRD analysis rule out the formation of TiS_2 after cycling. These results indicate the high electrochemical stability of CGPE. In S 2p XPS spectrum of cycled CGPE, the signals of LiPS, SO_3^{2-} and SO_2F^- are attributed to the residue of LiPS, electrolyte and their oxides in air. We also try to use graphite oxide (GO) as a 2D filler to reinforce the CGPE but the product failed to survive on flame. Related data have been added as Supplementary Figure 24 and Supplementary Figure 25 in the revised Supplementary Information with necessary discussion in the revised MS, page 15, highlighted in yellow. We also thank the reviewer to bring the above excellent works to our attention, which have been cited as Ref. 17 – 21 to enrich our revised MS.

Figure 4. (A) SEM image (B) EDS mapping of CGPE after cycling. (C) XRD patterns of CGPE before and after cycling.

Figure 5. XPS spectra of CGPE (A, B) before and (C, D) after cycling.

Q3. Why do plating is more uniform with GPE with respect to LE? Is it related to Li^+ ion transport number? Some review articles should be cited: a) *Angew. Chem. Int. Ed.* 55, 2016, 500–513; b) *Materials Science and Engineering: R: Reports* 134, 2018, 1-21; c) *Journal of Energy Chemistry* 56 (2021) 420–437;

Our Response: We thank the reviewer to inspire us about the positive effect of Li^+ transport number on Li plating and bring the above fantastic works to our attention, which have been cited as Ref. 46–48 in the revised MS.

The CGPE with high mechanical robustness could direct smooth Li plating on the lateral direction by inherence effect to form a Li layer with a smooth mosaic surface on Cu in contrast to mossy growth in liquid electrolyte. Meanwhile, the CGPE with a higher Li^+ transport number ($t_{\text{Li}^+} = 0.57$) than liquid

electrolyte (0.47) also facilitates uniform Li plating according to Sand's law, as suggested by the reviewer. Related data have been added as Supplementary Figure 20 in the revised Supplementary Information with necessary discussion in the revised MS, page 12 – 13, highlighted in yellow.

Figure 6. Polarization curves and impedance diagram (the inset) of (A) $\text{Li}_2\text{S}@\text{MX}||\text{LE}||\text{Cu}$ and (B) $\text{Li}_2\text{S}@\text{MX}||\text{CGPE}||\text{Cu}$ full cells before and after DC polarization. The t_{Li^+} is calculated by the Bruce-Vincent-Evans equation.

Q4. What is the role of 2% LiNO_3 ? When Li metal is used, LiNO_3 oxidize Li forming LiNO_x . What happens when Cu metal is used? The authors should characterize by XPS the Cu foil after lithium metal plating and define a mechanism.

Our Response: We thank the reviewer to point this out. The LiNO_3 is only added into the liquid electrolyte in half cells for stabilizing Li metal electrode and reducing LiPS shuttling, as extensively demonstrated in Li-S batteries (e.g., *Nat. Commun.* 2015, 6, 7436; *Chem* 2020, 6, 2533; *Angew. Chem. Int. Ed.* 2022, e202201406; *Electrochimica Acta* 2012, 70, 344). We did not use the LiNO_3 in anode-free $\text{Li}_2\text{S}@\text{MX}||\text{CGPE}||\text{Cu}$ cells. Therefore, there's no need to consider its effect on Cu metal.

Q5. As comparison the authors should test nickel metal foil instead of copper by using same different mass loading as shown in Figure 5b.

Our Response: We thank the reviewer for the good suggestion. We have tested the anode-free cells using Li_2S cathode with various mass loading of 5.0 – 15.2 mg cm^{-2} against Ni foil in CGPE ($\text{Li}_2\text{S}@\text{MX}||\text{CGPE}||\text{Ni}$). The CGPE also works effectively in such cells to deliver high capacities of 605 – 826 mAh g^{-1} with capacity retention of 60 – 65 % at a current density of 233.2 mA g^{-1} (0.2 C). Overall, the quasi-solid-state anode-free cells using Ni and Cu foils exhibit comparable performance. Related data have been added as Supplementary Figure 27 in the revised Supplementary Information with necessary discussion in the revised MS, page 15, highlighted in yellow.

Figure 7. Cycling stability of (A) $\text{Li}_2\text{S@MX}||\text{CGPE}||\text{Ni}$ and (B) $\text{Li}_2\text{S@MX}||\text{CGPE}||\text{Cu}$ cells with different Li_2S loadings at a current density of 233.2 mA g^{-1} (0.2 C).

Q6. As well the authors should report half-cell in manuscript. In Figure: S5 and S6 what is the loading used? The authors should test cathode with different loadings. Moreover, the final results should be compared with data cycling in Figure 5: it seems that the cells with a Cu anode show a better coulombic efficiency than half cells: why? In theory the results should be the opposite.

Our Response: We thank the reviewer for the excellent suggestion. The mass loading of Li_2S cathode for the tests in Supplementary Figure 5 and Supplementary Figure 6 is *ca.* 5.0 mg cm^{-2} , which has been clarified in the revised Supplementary Information, highlighted in yellow.

We have evaluated the cycling stability of $\text{Li}_2\text{S@MX}$ cathode with various mass loading of 1.5 and 3.0 mg cm^{-2} in half cells. In both cases, high capacities of $883 - 900 \text{ mAh g}^{-1}$ are achieved with over 90% capacity retention and high CE after 250 cycles at a current density of 233.2 mA g^{-1} (0.2 C). When Li_2S loading is increased to 9.5 mg cm^{-2} , the cells encountered short circuit by uncontrolled Li dendrite growth in liquid electrolyte (LE). Related data have been added as Supplementary Figure 6 in the revised Supplementary Information with necessary discussion in the revised MS, page 8, highlighted in yellow.

Figure 8. Cycling stability of $\text{Li}_2\text{S@MX}$ cathodes with mass loading of 1.5 and 3.0 mg cm^{-2} at a current density of 233.2 mA g^{-1} (0.2 C) in half cells.

Better CE of anode-free $\text{Li}_2\text{S}||\text{CGPE}||\text{Cu}$ cells relative to $\text{Li}_2\text{S}||\text{LE}||\text{Li}$ half cells is due to the positive effect of CGPE on good polysulfide retention and smooth Li plating. The CE of $\text{Li}_2\text{S}||\text{LE}||\text{Li}$ half cells are limited by Li_2S cathode due to overuse of far excess Li, which is mainly deteriorated by irreversible polysulfides loss from Li_2S cathode to liquid electrolyte. The reversibility of half cells can be improved by using CGPE to inhibit polysulfide loss, manifested as better CE of $\text{Li}_2\text{S}||\text{CGPE}||\text{Li}$ relative to $\text{Li}_2\text{S}||\text{LE}||\text{Li}$ cells. In anode-free $\text{Li}_2\text{S}||\text{CGPE}||\text{Cu}$ cells, the CE is firstly limited by sulfur inventory since initial irreversible polysulfide loss leaves excess Li. The CGPE also effectively improves the CE of anode-free cells at this stage likewise in half cells. After a period of cycling, irreversible Li loss will eventually govern the CE of anode-free cells and make it worse than half cells. Such undesired transition has been observed for anode-free cells using intercalation cathodes with intrinsic irreversible capacity loss (e.g., *Nat. Commun.* 2021, 12, 1452). Fortunately, it is largely delayed in our quasi-solid-state anode-free cells to secure high CE for over 300 cycles. Related data have been added as Supplementary Figure 26 in the revised Supplementary Information with necessary discussion in the revised MS, page 15, highlighted in yellow.

Figure 9. A comparison of $\text{Li}_2\text{S}@\text{MX}||\text{LE}||\text{Li}$ and $\text{Li}_2\text{S}@\text{MX}||\text{CGPE}||\text{Li}$ half-cell, as well as anode-free $\text{Li}_2\text{S}@\text{MX}||\text{CGPE}||\text{Cu}$ full cell in cycling stability and CE at a current density of 233.2 mA g^{-1} (0.2 C).

Q7. What is the mechanism of polysulfide retention? The authors should add a schematic. Probably the active species are Li_2S_6 and Li_2S_4 . The following reports should be used as reference a) *Nanoscale*, 2016,8, 13638-13645; b) *Sustainable Energy & Fuels*, 2017, 1 (4), 737-747; c) *Journal of Energy Chemistry* 54, 2021, 434–451; d) *Journal of Power Sources*, 2016, 319, 247-254.

Our Response: We thank the reviewer for the valuable suggestion. Electrochemical results and *in-operando* UV-vis analysis in our original MS have suggested the critical role of $\text{Ti}_3\text{C}_2\text{T}_x$ MXene in improving LiPS retention. On this basis, we further investigated the chemical interaction between LiPS and such MXene by XPS analysis of cycled $\text{Li}_2\text{S}@\text{MX}$ cathode. It reveals significant Lewis acid-base interaction between $\text{Ti}_3\text{C}_2\text{T}_x$ MXene and LiPS by Ti-S signals at 455.6/461.4 eV and 161.2/162.3 eV in Ti 2p and S 2p spectra, respectively. The presence of sulfite (167.3 eV), thiosulfate (168.8 eV) and trace polythionates (170.4 eV) also suggest the LiPS interaction with oxygen-containing groups on

MXene. Such chemical interactions effectively trap LiPS on the electrode interface, which not only inhibits LiPS loss to electrolyte but also promotes Li_2S dissolution and electrode kinetics with LiPS as redox mediators. Related data and schematic image have been added as Supplementary Figure 8 in the revised Supplementary information with necessary discussion in the revised MS, page 9, highlighted in yellow. We also appreciate the reviewer bringing the above illuminating works to enrich our work, which have been cited as Ref. 36 – 39 to enrich our revised MS.

Figure 10. (A) Ti 2p and (B) S 2p XPS spectra of cycled $\text{Li}_2\text{S}@MX$ cathode. (C) A schematic illustration of the positive role of MXene in promoting the reversibility and kinetics of Li-S redox conversion.

Reviewer # 2

This manuscript demonstrated an anode-free gel-electrolyte Li-S batteries using a Li₂S/MXene composite cathode. The key points of the study including (1) MXene is a superior conductive additive (than carbon) for Li₂S cathode and good catalysts for Li₂S oxidation and the sulfur reduction and (2) semi-solid electrolyte made of PVDF-HFP incorporated with LiTFSI, LiTFSI-based DOL/DME electrolyte, and small amount MXene can enable efficient Li deposition-stripping thus bring anode free. However, the mechanisms are not new, MXene catalysts for sulfur cathode, gel polymer electrolyte, and anode-free concept have all be reported before. Therefore, this manuscript should be submitted to a different journal with more specific scope.

Furthermore, some results in the current manuscript are vague:

Our Response: We thank the reviewer for the comment. Our work mainly focuses on developing fireproof quasi-solid-state anode-free lithium batteries with superior safety and volumetric energy, aiming to satisfy the urgent desire of high-energy yet reliable power sources. This new type of anode-free cell uses Li₂S cathode to replace commercial cathodes with 4 – 5 folds lower capacities. It enables not only high specific energy (> 300 Wh kg⁻¹) but also exceptional volumetric energy (> 1300 Wh L⁻¹) far beyond the current Li-ion batteries (< 750 Wh L⁻¹) and Li-metal batteries (< 500 Wh L⁻¹). More attractively, it secures remarkable safety against extreme use of short-circuits, overheating and even mechanical damage on fire, which is hardly achieved by common anode-free cells. These merits offer the opportunities to break the bottleneck of the existing power sources in energy density and safety, which determines the significant novelty of our work.

We fully agree with the reviewer that MXene has been used for catalyzing sulfur cathode. In our work, however, the MXene plays a distinct role in overcoming the huge charge potential barrier of bulk Li₂S, which is absent for sulfur. We also witnessed the success of various gel polymer electrolytes (GPEs). But they are still far from satisfactory in both performance and reliability by far. Our effort of utilizing the MXene to develop a new type of GPE with high ionic conductivity, thermal stability, wide electrochemical stability window and fire retardancy would be an important addition to this field. As for the mechanism, we uncovered the significant role of MXene in promoting LiPS-intermediated redox in quasi-solid-state anode-free cells by combined *in-operando* analysis for the first time. All these innovations can substantiate our claims in novelty again.

Q1. Figure 2F displays the calculated energy profile for Li₂S dissociation, but it is unclear why the reaction is Li₂S to LiS + Li: LiS does not exist.

Our Response: We thank the reviewer to point this out. This equation describes the primary step of Li₂S dissociation by breaking Li-S bond, which has been extensively used for calculating Li₂S dissociation in literature (*e.g.*, *PNAS* 2017, 114, 840; *J. Am. Chem. Soc.* 2019, 141, 3977; *Nat. Catal.* 2020, 3, 762; *Nat. Commun.* 2015, 6, 7760).

Q2. Mixing MXene into the polymer electrolyte (even with a low content) would increase the electronic conductivity of the electrolyte, which is not ideal.

Our Response: We thank the reviewer to remind us. Adding 3 % MXene into our polymer electrolyte induces a slight rise of electronic conductivity to $9.3 \times 10^{-11} \text{ S cm}^{-1}$ from $1.0 \times 10^{-11} \text{ S cm}^{-1}$ for MXene-free one at room temperature. It is much lower than the value of extensively used LLZO ($\sim 10^{-8} \text{ S cm}^{-1}$) and Li_3PS_4 ($\sim 10^{-9} \text{ S cm}^{-1}$) electrolyte, and is sufficient for dendrite-free Li plating (*Nat. Energy* 2019, 4, 187). Related data have been added as Supplementary Figure 14 in the revised Supplementary Information with necessary discussion in the revised MS, page 11, highlighted in yellow.

Figure 1. Current-time curves of the Cu||CGPE||Cu and Cu||MXene-free GPE||Cu cells under DC polarization at 100 mV.

Q3. The combination of the Li_2S loading and the Li_2S specific capacity is low in the full cells. It is unlikely the full cell can deliver a specific energy of 364 Wh/kg. It is very vague how this value was calculated. Same goes with the calculated 1363 Wh/L energy density. The authors must be very specific and detailed on the calculation of these numbers.

Our Response: We thank the reviewer for the good suggestion. The details in the calculation of gravimetric and volumetric energy of our cells have been added as Supplementary Table 1 in the revised Supplementary Information to support our claim.

Table 1. The calculation basis for cell energy of anode-free $\text{Li}_2\text{S}@MX||CGPE||Cu$ cells.

	Active mass (mg cm^{-2})	9.8
Cathode	Total mass (mg cm^{-2})	14.0
	Thickness (μm)	77.8
Electrolyte	Mass (mg cm^{-2}) ^a	20.2
	Thickness (μm)	20.0
Cu foil	Total mass (mg cm^{-2})	4.15
	Thickness (μm)	4.5
Cell	Cell mass (mg cm^{-2})	38.35
	Cell thickness (μm)	102.3

Average discharge voltage (V)	2.08
Gravimetric energy based on active mass (Wh kg⁻¹)	1423
Gravimetric energy on cell level (Wh kg⁻¹)^b	364
Volumetric energy based on active mass (Wh L⁻¹)	2376
Volumetric energy on cell level (Wh L⁻¹)^b	1363

^a 1.5 $\mu\text{L mgLi}_2\text{S}^{-1}$ LE is added.

^b The specific and volumetric energy on cell level are calculated based on the total weight and thickness of the cathode, electrolyte and Cu foil without including the package.

Q4. Similarly, the size of the full cells in this work seems very small, therefore the comparisons of specific energy and energy density in Figure 5c and 5d are incorrect. Because at least some of the values from the cited studies are from high capacity practical pouch cells (such as Ref 45).

Our Response: We thank the reviewer for the very constructive suggestion. For better comparison, we have assembled larger 0.51 Ah pouch-type anode-free cells, which have > 2 times higher capacity than that of practical pouch cells (0.23 Ah) in Ref. 45. Such cells consist of 4 layers of commercial Cu foil sandwiched by 3 layers of Li₂S cathodes and gel polymer electrolyte. High specific energy over 340 Wh kg⁻¹ and volumetric energy over 1320 Wh L⁻¹ are achieved to excel most reported anode-free cells and the state-of-the-art Li-ion and Li-S batteries. The details in the calculation of cell energy have been added as Supplementary Table 2 in the revised Supplementary Information to support our claim. Related data have been as Figure 5C in the revised MS and Supplementary Figure 31 in the revised Supplementary Information with necessary discussion in the revised MS, page 16. Figure 5C and Figure 5D in the original MS have been revised to Figure 5D and Figure 5E by adding the data of our 0.51 Ah pouch-type anode-free cells. All the changes have been highlighted in yellow.

Table 2. The calculation basis for cell energy of 0.51 Ah Li₂S@MX||CGPE||Cu cells.

	Active mass (mg cm ⁻²)	4.8
Cathode	Area (cm ⁻²)	22.5
	Total mass (mg)	925.0
	Total thickness (μm)	228.6
	Total mass (mg) ^a	1880.0
Electrolyte	Total thickness (μm)	120.0
	Area (cm ⁻²)	24.44
Cu foil	Total mass (mg)	406.0
	Total thickness (μm)	18.0
Cell	Capacity (Ah)	0.51
	Cell mass (mg)	3211

Cell thickness (μm)	366.6
Average discharge voltage (V)	2.13
Gravimetric energy based on active mass (Wh kg^{-1})	1684
Gravimetric energy on cell level (Wh kg^{-1}) ^b	340
Volumetric energy based on active mass (Wh L^{-1})	2797
Volumetric energy on cell level (Wh L^{-1}) ^b	1323

^a1.5 $\mu\text{L mgLi}_2\text{S}^{-1}$ LE is added.

^bThe specific and volumetric energy on cell level are calculated based on the total weight and thickness of the cathode, electrolyte and Cu foil without including the package.

Figure 2. (A) Discharge-charge voltage curves and (B) cycling performance of a 0.51 Ah pouch-type $\text{Li}_2\text{S@MX}||\text{CGPE}||\text{Cu}$ cell.

Reviewer # 3

The surging development of electric vehicles and portable electronic devices eagerly desires powerful but thinner batteries that can store more energy in limited space with high reliability. The quasi-solid-state anode-free batteries developed in this work perfectly meet this desire due to impressive volumetric energy and safety against abuse. Stable redox of ultrahigh-loading Li_2S cathode in fireproof gel polymer electrolyte offers new opportunities of harvesting both excellent volumetric energy and safety. Direct use of commercial bulk Li_2S without tedious material fabrication further adds the practical value. The strategy applying robust polymer electrolyte to zero-excess Li metal also inspires one to address the Li dendrite issues in Li metal batteries. Such an innovative cell design holds promise in not only developing thinner cells but also breaking the safety bottleneck threatening the practical use of alkali-metal based batteries. I encourage the publication of this work in Nature Communications with a necessary revision of several points.

Our Response: We are grateful to the reviewer for high recognition of our work and very constructive suggestions.

Q1. More evidence such as CVs is necessary to clarify the reaction pathway of anode-free $\text{Li}_2\text{S}||\text{Cu}$ full cells.

Our Response: We thank the reviewer for the good suggestion. The CVs of anode-free $\text{Li}_2\text{S}@\text{MX}||\text{CGPE}||\text{Cu}$ cells have been measured at a scan rate of 0.1 mV s^{-1} . Such cells show an anodic peak at *ca.* 2.54 V for Li_2S activation during the initial CV scan from the open-circuit voltage (OCV, *ca.* 0.3 V) to 3.5 V. The rest of the cycles exhibit highly overlapping CVs with an anodic peak at *ca.* 2.4 V for Li_2S oxidation to sulfur and two cathodic peaks at *ca.* 2.3 and 2.0 V for sulfur reduction to LiPS and finally to Li_2S , suggesting a similar reversible LiPS-intermediated redox pathway with that in half cells. Related data have been added as Supplementary Figure 21 in the revised Supplementary Information with necessary discussion in the revised MS, page 14, highlighted in yellow.

Figure 1. CVs of anode-free $\text{Li}_2\text{S}@\text{MX}||\text{CGPE}||\text{Cu}$ cell at a scan rates of 0.1 mV s^{-1} .

Q2. It has been reported that the Li_2S cathode can be paired with high-capacity anodes such as Si for manufacturing high-energy batteries. Can the author comment on the potential of this kind of battery?

Our Response: As mentioned by the reviewer, the battery made of Li_2S cathode and high-capacity Si anode is promising in harvesting high theoretical energy of 1550 Wh kg^{-1} from the multi-electron involved redox reactions. Meanwhile, this cell chemistry also secures intrinsic high safety by avoiding uncontrolled exothermic chain reactions of extremely reactive species such as Li metal and radical oxygen. These benefits make such cells attractive to the urgent demand of high-energy yet reliable power sources beyond the state-of-the-art Li-ion batteries. Nevertheless, the use of Si anode raises the processing complexity and cost of cell production while inevitably sacrificing the cell energy by its mass and volume occupied. This bottleneck motivated us to develop the anode-free battery to take the essence of Li_2S cathode without the drawback of Si anode for maximizing the cell energy, as demonstrated in the present work.

Q3. What is the interaction between LiPS and MXene? Please specify to avoid any misunderstanding.

Our Response: We thank the reviewer for the good suggestion. XPS analysis of cycled $\text{Li}_2\text{S}@MX$ cathode reveals significant Lewis acid-base interaction between $\text{Ti}_3\text{C}_2\text{T}_x$ MXene and LiPS by Ti-S signals at 455.6/461.4 eV and 161.2/162.3 eV in Ti 2p and S 2p spectra, respectively. The presence of sulfite (167.3 eV), thiosulfate (168.8 eV) and trace polythionates (170.4 eV) also suggest the LiPS interaction with oxygen-containing groups on MXene. Such chemical interactions effectively trap LiPS on the electrode interface, which not only inhibits LiPS loss to electrolyte but also promotes Li_2S dissolution and electrode kinetics with LiPS as redox mediators. Related data have been added as Supplementary Figure 8 in the revised Supplementary information with necessary discussion in the revised MS, page 9, highlighted in yellow.

Figure 2. (A) Ti 2p and (B) S 2p XPS spectra of cycled $\text{Li}_2\text{S}@MX$ cathode.

Q4. In Figure 5B, the discharge capacity of full cells with higher Li₂S loading increases in the initial cycles. What is the reason?

Our Response: We thank the reviewer to point this out. Such capacity rise during initial cycles is a result of the gradual activation of micrometer-sized Li₂S in thick electrodes with very high mass loading (*e.g.*, 14 mg cm⁻²) in gel polymer electrolyte. Related discussion has been added in the revised MS, page 14 – 15, highlighted in yellow

Q5. Please provide the ionic conductivity of CGPE at different temperatures.

Our Response: We are grateful to the reviewer for the excellent suggestion. We have measured the ionic conductivity of CGPE at various temperatures. It rises from 0.8 to 1.78 mS cm⁻¹ with temperature increasing from 298.15 to 378.15 K according to the Arrhenius model. This result suggests the ionic transport in CGPE undergoes a rafting process decoupled from the long-range motion of polymer chains. Related data has been added as Supplementary Figure 13C in the revised Supplementary Information with necessary discussion in the revised MS, page 11, highlighted in yellow

Figure 3. Temperature-dependent ionic conductivity of CGPE. The solid line is fitted by using the Arrhenius transport model.

Q6. The label of the y-axis in Figure 2F may be “reaction pathway”. It would be better to just present the reaction equation here for better readability.

Our Response: We thank the reviewer to point this out. The label of the y-axis in Figure 2F has been revised to “Li₂S → LiS + Li⁺ + e⁻” for better readability.

REVIEWERS' COMMENTS

Reviewer #1 (Remarks to the Author):

The authors have well revised the manuscript: therefore I recommend to accept the paper in the present form

Reviewer #2 (Remarks to the Author):

The authors fully addressed the comments from the reviewers in their revision. The current version can be accepted for publishing.

Reviewer #3 (Remarks to the Author):

I think the authors have addressed all my concerns, thus I believe it is time to recommend its publication in Nature Communications.

Reviewer # 1

The authors have well revised the manuscript: therefore I recommend to accept the paper in the present form.

Our Response: We highly appreciate the reviewer for his time and very valuable comments in improving the quality of this manuscript again.

Reviewer #2

The authors fully addressed the comments from the reviewers in their revision. The current version can be accepted for publishing.

Our Response: We highly appreciate the reviewer for his time and very valuable comments in improving the quality of this manuscript again.

Reviewer #3

I think the authors have addressed all my concerns, thus I believe it is time to recommend its publication in Nature Communications.

Our Response: We highly appreciate the reviewer for his time and very valuable comments in improving the quality of this manuscript again.